



# The optimum range of design axial induction factors for lowest levelized-cost of energy

Desirae Major and Sven Schmitz[1]

[1]The Pennsylvania State University, University Park, PA, 16801

**Correspondence:** Desirae Major (dzm305@psu.edu)

**Abstract.** The present work proposes a new fatigue, aerodynamics, and cost-scaled turbine (FACT) blade design methodology. The basis of the FACT blade design is an objective function for change in levelized-cost of energy, $\Delta LCOE$, as a function of design axial induction factor, $a$, that strikes a balance between increasing (or up-scaling) blade length and annual energy production while accounting for the additional cost and loading changes associated with a longer blade. In the process of developing the $\Delta LCOE$ objective function, new insights were gained about changes in capital cost and operations and maintenance cost with rotor up-scaling. As part of the capital cost function development, new engineering approximations for rotor mass are discussed that are suitable for large-diameter offshore wind turbines, which use improved materials technologies and manufacturing processes. Additionally, a detailed operations and maintenance model is developed using available wind farm reliability data. Furthermore, a relationship between turbine failure rate and damage-equivalent loads for failure-prone turbine subsystems is proposed. FACT rotor blade design points are identified using five reference wind turbines with power ratings of 10- to 22-MW as a baseline. Projected LCOE savings with a FACT rotor blade design are on the order of 5% for an optimum design axial induction factor in the range of $a = 0.21$ and 0.27, thus falling between the low-induction rotor concept ($a = 0.18$) and the Betz optimum for maximum $C_P$ ($a = 0.33$).

## 1 Introduction

Current pathways aim to limit the global average temperature increase to no more than 2 °C by 2100, with net-zero greenhouse gas emissions achieved by 2050 (IPCC, 2019). Wind energy is predicted to play a substantial role in the electrification and decarbonization of the energy sector toward achieving net-zero emissions by 2050. The pathway to net-zero emissions estimates that low- and zero-emissions sources will contribute as much as 97% of the global electricity production in 2050 (IPCC, 2019). Wind and solar are predicted to be the primary renewable energy producers, contributing up to 50% of the global electricity consumed by 2050 (IEA, 2021). A significant gap remains, however, between the current growth rate of the wind industry and the growth targets required to achieve net-zero emissions by 2050.





One strategy to improve wind energy penetration in the global electricity market is increasing wind turbine diameters for offshore wind farms to capture as much energy as possible per turbine, i.e. up-scaling. Since 1999, wind turbine rotor diameters have grown at a rapid pace, increasing as much as 159% (Wiser et al., 2021). As offshore wind turbine diameters become larger, new scientific and engineering challenges arise that may hinder rapid implementation of these new technologies, including keeping rotor loads low to ensure a sufficient operational lifetime of the wind turbine and keeping cost of electricity competitive with traditional fossil fuel energy sources (Veers et al., 2022, 2023; Jensen et al., 2017).

One concern for the design of wind turbines is fatigue. Wind turbines are fatigue-prone machines, with some components experiencing as many as $10^9$ fatigue cycles during the 20 - 30 year design lifetime of the machine (Sutherland, 2002; Mandell et al., 1992; Nijssen, 2022). As wind turbine blades become longer, they become more flexible and traverse higher in the atmospheric surface layer during one rotation cycle, causing significant load variation over one blade passage (Veers et al., 2023). Additionally, as wind turbines are aero-elastic machines, the increased aerodynamic loads pose concerns for the stability of the elastic loads of the long, flexible blades of future wind turbines (Veers et al., 2022).

The cost of wind energy is given by the levelized-cost of energy (LCOE), which is the sum of the total costs to produce the energy divided by the annual energy produced (AEP). Though the LCOE of wind-produced electricity has dropped 45% over the last decade and wind energy is already competitive with traditional fossil fuels in some markets (IPCC, 2019), this is mostly true for electricity generated by onshore wind farms. The cost of offshore wind is notably more expensive, nearly twice that of onshore wind (Stehly et al., 2020), and market estimates predict that wind turbine LCOE will actually increase 9% in the near-term (GWEC, 2022).

With the capacity of the most profitable onshore wind farm sites already saturated and limited land availability to develop more onshore wind farms, more wind farms are expected to move offshore (Veers et al., 2023). As more wind farms move off-shore, wind turbine size can also increase due to less stringent noise requirements, thereby increasing annual energy production per machine, which lowers overall LCOE. However, the increasing turbine size also increases the total cost of the machine. Total cost in the LCOE equation is a sum of the capital cost (CAPEX), or the initial investment to manufacture and install the machine, plus the operations and maintenance cost (OPEX). As wind turbines grow larger to meet global electrification goals, wind turbine capital cost is also expected to increase. Longer blades means higher materials and manufacturing costs for the rotor, plus increased cost for the increased generator size to match the increased power production requirement, and increased tower and foundation costs as more substantial foundations are required to support the large rotors and nacelles.

Besides CAPEX increasing as turbines grow larger, OPEX costs are also expected to substantially increase. OPEX costs of offshore wind farms are already nearly twice that of onshore wind farms due to different operating regimes, environmental conditions, and larger turbines (Bakhshi and Sandborn, 2018; Dao et al., 2019; Li et al., 2020; Ozturk et al., 2018; Stehly et al., 2020). Additionally, increased blade loading and LCOE are a coupled problem through the OPEX component of the total cost. Longer and more flexible blades mean significant changes to the aeroelastic behavior of the rotor and related mechanical subsystems that must be designed to handle the rotor loads. These increased loads may excite new failure modes in the turbine structure, causing changes to turbine failure rates and needs for different and more costly maintenance strategies.



From *Grand challenges in the design, manufacture, and operation of future wind turbine systems*, the authors note that tackling these challenges of the future of wind energy will require new design methodologies (Veers et al., 2023). Many recent efforts within the community have sought to address this goal and focused on developing holistic design strategies using a variety of computational methods. Ashuri et al. (2014) developed and investigated a multi-disciplinary optimization (MDO) joint optimization routine to design a rotor and tower for minimum LCOE. It employed extensive cost modeling using the relationships from Fingersh et al. (2006) to estimate CAPEX and OPEX changes as a result of the new turbine design. At most, the model yielded a 2.3% decrease in LCOE for a re-designed NREL 5-MW turbine. The LCOE optimum in this scenario still increased CAPEX and OPEX costs, which were ultimately compensated for by an increase in AEP.

Though not a study on economic impacts of an optimum design, the work of Zalkind et al. (2019) focused on system design for rotors of 100-m and longer radii such that AEP is maximized while constraining overall turbine loads with concern to OPEX and CAPEX increases. For blade design, the framework investigated variations in blade length, cone angle, number of blades, rotor axial induction factor, as well as rotor mass and stiffness distribution to determine the optimum configuration. Studies were also conducted on the hub configuration, nacelle layout, and tower design to look at turbine design from a system perspective. From this study, the authors estimated that an 11% increase in AEP for a 13-MW wind turbine could be achieved while successfully constraining rotor loads.

The work of Hietanen et al. (2023) focused on optimizing the layout of an offshore wind farm, specifically with the goal of reducing LCOE of offshore wind farms that use novel floating wind turbines versus traditional fixed-bottom wind turbines. In general, floating wind turbines are more costly than their fixed-bottom counterparts, but floating substructures allow installation of wind farms in deeper waters. The optimization routine in Hietanen et al. (2023) determines rotor placement in various offshore wind farms using an objective function that seeks to maximize AEP against CAPEX and OPEX constraints. The focus of this optimization with rotor placement is not the wind turbine design, but rather the rotor installation and transmission requirements, which are significant costs for floating offshore wind farms.

Returning to the wind turbine rotor design, the work by Canet et al. (2023) represents a new design perspective that does not focus purely on power production or economic metrics, but rather investigates long-term sustainability and lifetime $CO_2$ emissions of a wind turbine. This work developed a relationship for $CO_2$ emissions equivalent to the LCOE relationship: environmental value minus the environmental cost over the total energy produced. Using the new metric, hub height and rotor diameter were optimized to minimize the environmental impact. Results found that the environmental optimum is not the LCOE optimum, and some compromise must be found between the two optima for long-term sustainability.

One final work that focuses on economic optimization along with AEP optimization is that of Buck and Garvey (2015). Here, the authors note that CAPEX is the most significant contributor to the total turbine cost in the numerator of the LCOE equation. The routine developed in this work optimizes the blade chord, axial induction factor, and sectional lift coefficient along the span of a blade to minimize CAPEX while maximizing AEP. Using the NREL 5-MW as a baseline, the planform was optimized to reduce CAPEX by 1%, increase AEP by 1.4%, for an overall reduction of the CAPEX/AEP ratio of 2.4%.

In line with these efforts, the present work seeks to add to the literature on optimizing wind turbine design with respect to LCOE. The novelty in the present work is the focus on minimizing the LCOE of an up-scaled rotor. While up-scaling is gener-





ally understood to be a successful way to increase power production (Papi and Bianchini, 2022), it can lead to increased costs that negate the cost benefit of the increased AEP. The effect of up-scaling on cost, and particularly OPEX, is less understood (Papi and Bianchini, 2022). Among the Grand Challenges (Veers et al., 2023), it was also identified that the cost of up-scaling

offshore rotors should be addressed. It should also be noted that up-scaling is not meant here in the traditional sense where a 10-MW rotor is scaled up to 15-MW, but rather that the rotor diameter of the 10-MW rotor is up-scaled just slightly to increase Region II energy capture and increase overall AEP while maintaining the same rated capacity. This is an effort to make each turbine deployed into wind farms as efficient as possible and the energy produced as cheap as possible for a given rated power.

The rotor design parameter of interest to optimize for the up-scaled rotor is the average design axial induction factor, $a$.

Choosing the axial induction factor as the parameter to optimize is ideal as it gives a direct first-order measure of rotor loads and power production. From momentum theory, rotor thrust coefficient ($C_T$) is approximated by $C_T = 4a(1-a)$ and rotor power coefficient ($C_P$) is $C_P = 4a(1-a)^2$. Additionally, it has been shown that for a given design axial induction factor, an optimum spanwise circulation distribution can be determined using a vortex wake method (VWM), from which the rotor planform can be quickly designed given either a spanwise chord, twist, or lift coefficient distribution constraint (Major et al.,

2022; Schmitz, 2020).

The authors first investigated the initial concept of up-scaling a baseline rotor design based on a design axial induction factor using the low-induction rotor concept in Major et al. (2022). The low-induction objective function was shown to be optimized for $a = 0.18$, resulting in a blade that is 13% longer than the reference rotor and yields more than a 6% increase in AEP and a nearly 4% decrease LCOE (Major et al., 2022). When the cost of the proposed low-induction concept was further investigated

using the tools available, a notable increase in CAPEX was observed relative to the reference rotor from which the up-scaled rotor was designed and that OPEX was not a factor in the cost model due to lack of a sufficient OPEX cost model.

Based on the results of this previous work by the authors here, the hypothesis is that there is an optimum average axial induction factor at which an up-scaled rotor blade can be designed that strikes the right balance between increasing a blade length and AEP while accounting for the additional cost and fatigue loading changes associated with a longer blade. Thus,

the goal of the present work is to develop a relationship of the following form: $\Delta LCOE = \Delta LCOE(a)$. Using $\Delta LCOE$, the present work seeks to investigate the potential change in LCOE for an up-scaled rotor relative to the baseline rotor from which it was designed. Breaking down LCOE further into its components, relationships for $\Delta CAPEX(a)$, $\Delta OPEX(a)$, and $\Delta AEP(a)$ are sought such that trends in each of these components with design axial induction factor, and thus blade length, can be also investigated and better understood for up-scaled rotors. The result is a design objective function equation of the

following form:

$$\Delta LCOE(a) = \frac{\Delta CAPEX(a) + \Delta OPEX(a)}{\Delta AEP_{net}(a)} \longrightarrow min < 0 \qquad (1)$$

To accomplish the goal of developing a function for $\Delta LCOE(a)$, the remainder of the present work is organized as follows: Section 2 details the development of the model for $\Delta CAPEX$ using the work of Buck and Garvey (2015) as a foundation, $\Delta OPEX$ model development in Section 3 including the relationship between blade loads and $\Delta OPEX$, and case studies





using the combined LCOE model to find the optimum range of design axial induction factors for a fatigue, aerodynamics, and cost-scaled turbine (FACT) rotor blade design in Section 4.

## 2  Capital Cost (CAPEX) Model

The first step of the present study was to develop a function for CAPEX in terms of design axial induction factor, $a$. The basis of the function comes from the work of Buck and Garvey (2015) where CAPEX optimization is governed by a reduction of the

total structural material and bending stresses (Buck and Garvey, 2015). Total CAPEX of a wind turbine is assumed to be the summation of the rotor, tower, and nacelle material cost plus other miscellaneous components and manufacturing costs (Buck and Garvey, 2015; Dykes et al., 2014). The result is the following simple formulation for the estimated change in CAPEX for a new rotor design, $\Delta CAPEX$, relative to a baseline rotor (denoted by the superscript "0") (Buck and Garvey, 2015)

$$\Delta CAPEX = A\left(\frac{M_{rotor}}{M_{rotor}^0}\right) + B\left(\frac{S_{tower}}{S_{tower}^0}\right) + C\left(\frac{Q_{shaft}}{Q_{shaft}^0}\right) + D \tag{2}$$

Here blade cost is proportional to mass of the rotor, $M_{rotor}$, tower cost is scaled by the tower structural requirements, $S_{tower}$, and the nacelle cost is scaled by the required drivetrain shaft torque, $Q_{shaft}$. To further expand these relationships, Buck and Garvey (2015) note that the tower structural capacity, $S_{tower}$, and shaft torque, $Q_{shaft}$, can be estimated by

$$S_{tower} = F_T\left(\frac{H^2 + W^2}{W}\right) \quad \text{and} \quad Q_{shaft} = \left(\frac{1}{\eta_{gen}}\right)\frac{P_{rated} * R}{u_{tip}} \tag{3}$$

where $F_T$ is the rotor aerodynamic thrust force, $H$ is the tower height, and $W$ is the tower base width, $\eta_{gen}$ is the generator

efficiency, $P_{rated}$ is the rated power output of the wind turbine, $R$ is blade radius, and $u_{tip}$ is the blade tip speed. The rotor mass relationship will take the form of $M_{rotor} = N * R^x$, which is a typical engineering approximation for rotor mass used in the community, and will be discussed in detail in Section 2.2. The coefficients $A, B, C$, and $D$ are the proportional contributions of each turbine component to the total capital cost and will be discussed in more detail in Section 2.3.

### 2.1  CAPEX Axial Induction Relationships

To determine the optimum range of design axial induction factor at which to design a blade to minimize LCOE, it is desirable to express Equation 2 as a function of the former. In the present work, this is achieved by coupling the equation to the low-induction objective function (Chaviaropoulos and Sieros, 2014) and defining relevant quantities in the equations for $S_{tower}$, $Q_{shaft}$, and $M_{rotor}$ as a function of axial induction factor using momentum theory principles. The LIR concept was identified as a novel technology with the potential to increase wind farm capacity factor while keeping levelized-cost of energy (LCOE)

competitive with current technologies, and studies have shown that it has potential to notably reduce blade and tower fatigue over the lifetime of the turbine (Major et al., 2022).



The low-induction rotor (LIR) seeks to maximize power ($C_P$) given a bending moment constraint ($\sim C_T^{3/2}$), or $C_p/C_T^{2/3} \rightarrow max$. This expression is optimized at $a = 0.18$, resulting in an optimum $C_T$ of 0.6 for a low-induction rotor (Schmitz, 2020). Blade radius, $R$, is a free variable in the LIR design and is scaled by the resulting design $C_T$ of the new low-induction rotor according to (Chaviaropoulos and Sieros, 2014):

$$\frac{R(a)}{R^0} = \left( \frac{C_T^0}{C_T(a)} \right)^{1/3} = \left( \frac{C_T^0}{4a(1-a)} \right)^{1/3} \tag{4}$$

Equation 4 is plotted in Figure 1. For generality, and to understand the trend of blade radius with axial induction factor, let's assume the Betz Optimum as the baseline rotor from which an up-scaled rotor would be designed: $C_T^0 = 0.89$ for $a = 1/3$. Note that the range of design axial induction factors over which the data are plotted is $a = [0.1, 0.35]$, where $a = 0.35$ is large enough to include the Betz optimum rotor axial induction factor. The lower bound of the design axial induction factor plot is limited to $a = 0.1$ because the relationship for $R(a)$ sees an exponential increase past this value, resulting in extremely large rotors that are undesirable for minimizing cost.

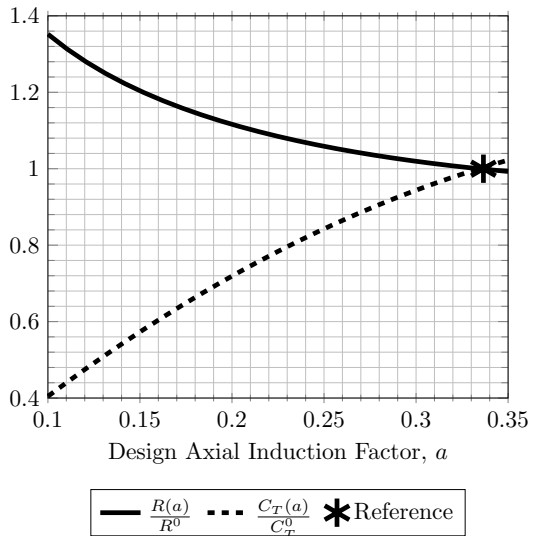

**Figure 1.** Blade radius and design trust coefficient as a function of design axial induction factor for the low-induction objective function with the Betz Optimum rotor as the baseline (0.33, 1).

As design axial induction factor decreases away from the Betz optimum ($a = 1/3$), blade radius of the new rotor design increases relative to the reference rotor size, allowing for the baseline rotor to be "up-scaled". This is a special result of the low-induction design constraint on the root flap bending moment, which results in the decreasing design thrust coefficient ratio also plotted in Figure 1.



Applying momentum theory approximations and using the new definition of rotor radius, $R = R(a)$, the rotor aerodynamic thrust force, $T$, that scales the tower structural cost can then be re-written as a function of axial induction factor according to

$$T(a) = 2\rho V_o^2 a (1-a) \pi R(a)^2 \tag{5}$$

where $\rho$ is density and $V_o$ is the design wind speed of the up-scaled rotor. Note that the structural requirements for tower width, $W$, and height, $H$, also scale with turbine size. Here the authors assume that the same tower dimensions that support the reference rotor can support the up-scaled rotor. This is a reasonable assumption because the rated power of the up-scaled rotor is unchanged, the low-induction bending moment constraint is employed and thus, the thrust force that the tower will be exposed to for the up-scaled rotor is lower (see Figure 1) while at the same time the up-scaled blade length is not significantly

larger than the baseline rotor to require a taller tower. With the definition of $T$ as a function of design axial induction factor, $a$, this gives the final form of the $S_{tower}$ term in Eq. (2) as follows:

$$S_{tower}(a) = T(a) \left( \frac{H^2 + W^2}{W} \right) \quad \text{where} \quad T(a) = 2\rho V_o^2 a (1-a) \pi R(a)^2 \tag{6}$$

Normalizing this relationship by the reference value, $S_{tower}^0 = \frac{1}{2}\rho C_T^0 V_o^2 \pi (R^0)^2 (\frac{H^2+W^2}{W})$, assuming that the design thrust of the up-scaled rotor occurs at the same design wind speed, $V_o$, and applying the assumption for the tower dimensions discussed

above yields the following relationship for $\Delta S_{tower}(a)$:

$$\Delta S_{tower}(a) = \frac{S_{tower}}{S_{tower}^0} = \frac{4a(1-a)}{C_T^0} \left( \frac{R(a)}{R^0} \right)^2 \tag{7}$$

Finally, the shaft torque, $Q_{shaft}$, is defined in terms of axial induction factor by noting that $R = R(a)$:

$$Q_{shaft}(a) = \frac{P_{rated} * R}{\eta_{gen} * u_{tip}} = \frac{P_{rated} * R(a)}{\eta_{gen} * V_o * TSR} \tag{8}$$

The generator efficiency, $\eta_{gen}$, is assumed to be 94% for all cases and the rated power, $P_{rated}$, is set by the design require-

ments. Note that $u_{tip}(= \Omega R)$ was replaced with $V_o * TSR$, where $TSR = \Omega R / V_o$ is the design tip speed ratio. This is done to emphasize that rotors are typically designed around a target design wind speed, $V_o$, and a design $TSR$. Normalizing the value of $Q_{shaft}$ for the up-scaled turbine by the reference value, $Q_{shaft}^0 = \frac{P_{rated} * R^0}{\eta_{gen} * V_o * TSR^0}$, gives the following relationship for $\Delta Q_{shaft}(a)$:

$$\Delta Q_{shaft}(a) = \frac{R(a)}{R^0} \tag{9}$$

In this simplification, the authors assume not only that the design wind speed, $V_o$, is the same between the reference and up-scaled rotor, but also that the rated power, $P_{rated}$, and design $TSR$ are the same. The goal of the up-scaling in the present



work is to up-scale a reference rotor to a size that produces more AEP for a given rated power and $TSR$. This relationship for $\Delta Q_{shaft}(a)$ would change if the method were used to design an up-scaled rotor for a different rated power.

Though the final form of $M_{rotor}$ is unknown, the general function for $M_{rotor}$ that will be used in the present work was previously discussed. Normalizing this relationship by the reference value yields:

$$\Delta M_{rotor}(a) = \left( \frac{R(a)}{R^0} \right)^X \tag{10}$$

The determination of the exponent $X$ for $\Delta M_{rotor}(a)$ is discussed in detail in the next section.

## 2.2 Rotor Mass Relationship

One unknown in Eq. (2) that requires additional investigation is the function for rotor mass, $M_{rotor}$. In the original work from which the $\Delta CAPEX$ function is adapted, $M_{rotor}$ is determined using an iterative process to minimize spar cap and skin material surface area for a given blade chord under a load constraint (Buck and Garvey, 2015). In the present work, a more traditional engineering approximation for rotor mass is developed based on rotor blade length, which allows rotor mass to be represented as a function of design axial induction factor. A well-known equation of this form is found in the work of Fingersh et al. (2006); however, this approximation was developed for onshore wind turbines (40-m $\leq R \leq$ 70-m) and older wind turbine material technologies than the high-tech, multi-megawatt offshore machines in production today.

Using the general form of the engineering approximation, $M_{rotor}(R) = M_{rotor}(a) = N * R(a)^X$, the first new relationship suitable for large-diameter offshore wind turbines is proposed using a combination of reference turbines and commercial wind turbines with rotor blade masses either determined through detailed structural layups (in the case of the reference turbines) or publicly available data (for the commercial wind turbines). The list of all turbines considered is given in Table 1.

**Table 1.** Reference and commercial turbines used to develop a relationship for blade mass as a function of blade radius, $R$.

| Rotor | Blade Radius | Blade Mass |
|---|---|---|
| NREL 5-MW (Jonkman et al., 2009) | 63.0 m | 17,740 kg |
| Vestas V164 (Vestas, 2023) | 80.0 m | 33,000 kg |
| LM Windpower 8-MW (Wittrup, 2016) | 88.4 m | 33,700 kg |
| DTU 10-MW (Bak et al., 2013) | 89.2 m | 40,814 kg |
| IEA 10-MW (Bortolotti et al., 2019) | 99.0 m | 47,700 kg |
| GE Haliade-X (GE, 2023) | 110.0 m | 55,000 kg |
| IEA 15-MW (Gaertner et al., 2020) | 120.0 m | 65,250 kg |
| UPWIND 20-MW (Peeringa et al., 2011) | 126.0 m | 59,253 kg |
| IEA 22-MW (Zahle et al., 2024) | 142.0 m | 83,342 kg |

The other relationship for blade mass as a function of rotor radius considered in the present paper comes from projections done by the blade manufacturing company LM Windpower shown in Figure 2, adapted from (Bottasso, 2019). This figure





shows various trends for blade mass for different material types assuming different degrees of technological advancement. In the present work, only LM Hybrid Carbon (maroon squares in Figure 2) was considered as it covers turbines with radii larger than 80-m, making this data representative of typical large offshore wind turbines.

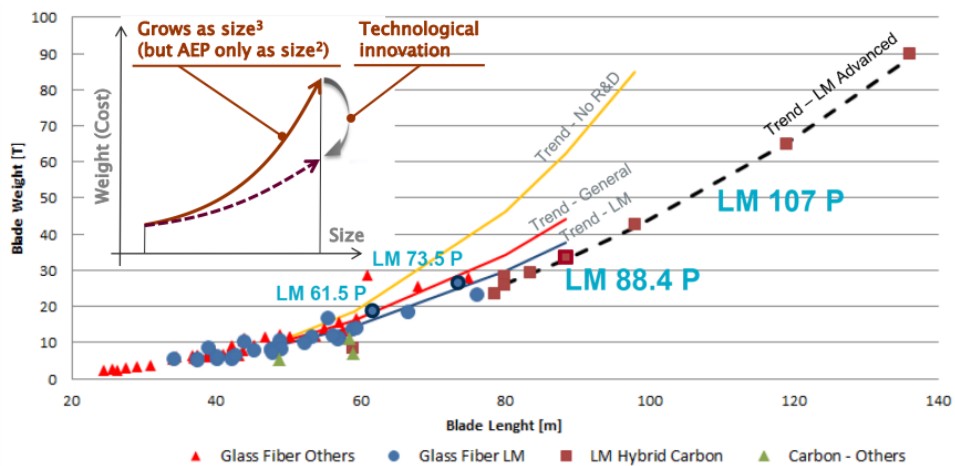

**Figure 2.** Relationships for blade mass as a function of blade radius for various LM Windpower technologies, adapted from (Bottasso, 2019).

The LM Hybrid Carbon data from Figure 2 and the data from Table 1 were plotted separately and a power curve was applied to determine the new functional engineering approximations for blade mass as a function of blade radius. The relationships are given in Table 2 along with the R-squared value representing goodness-of-fit and the Fingersh et al. relationship for comparison.

**Table 2.** Approximations for blade mass as a function of blade radius.

| Data | Equation | R-Squared |
|---|---|---|
| Fingersh et al. (2006) | $0.4948R^{2.53}$ | - |
| Reference + Commercial Turbines | $12.077R^{1.7863}$ | 0.9595 |
| LM Hybrid Carbon | $0.8520R^{2.3563}$ | 0.9988 |

The relationships in Table 2 are compared graphically in Figure 3. At lower blade radii in the range of 60-m, all three functions predict approximately the same blade mass. The two new relationships proposed here, however, show a significant
reduction in blade mass compared to the older Fingersh et al. relationship as blade radius increases to 140-m. Technological advancements in materials and blade structural optimization codes have improved material usage in the reference and commercial blades, which is reflected in the newer relationships proposed here. Comparing the LM Hybrid Carbon and the Reference + Commercial Turbine curves, the results are relatively consistent over the range of blade lengths considered.





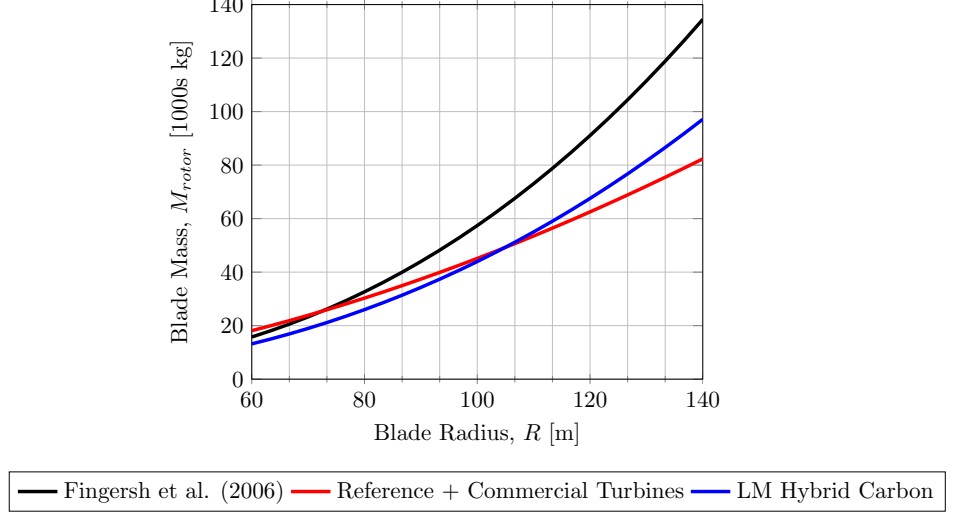

**Figure 3.** Predictions for blade mass, $M_{rotor}$, as a function of blade radius, $R(a)$, for various material technologies and turbine sizes.

## 2.3 Component Contribution to Change in Total CAPEX

For the $\Delta CAPEX$ relationship published in the work of Buck and Garvey (2015) and used as the starting formulation here, the constants $A$, $B$, $C$, and $D$ in Eq. (2) are based on the work of Krohn, Morthorst and Awerbuch (Krohn et al., 2009). This work developed the proportionality constants using data for the NREL 5-MW and result in the following first formulation for $\Delta CAPEX$:

$$\Delta CAPEX = 0.2\left(\Delta M_{rotor}\right) + 0.2\left(\Delta S_{tower}\right) + 0.2\left(\Delta Q_{shaft}\right) + 0.4 \qquad (11)$$

Similar to the rotor mass relationship, this CAPEX proportionality relationship is based on older turbine technologies and smaller offshore wind turbines. To verify these proportionality constants, a study was conducted using four reference wind turbine designs of 10-MW rated power and higher to determine if the relative contribution of each component to the total CAPEX had changed with increasing turbine size and new turbine technologies. Absolute CAPEX for the reference turbines was estimated using the open-source tool WISDEM (Wind-plant Integrated System Design and Engineering Model) (Dykes

et al., 2014), which implements advanced cost models based on modern materials and manufacturing techniques. The four reference turbines used for the investigation were the DTU 10-MW (Bak et al., 2013), IEA 10-MW (Bortolotti et al., 2019), IEA 15-MW (Gaertner et al., 2020) and IEA 22-MW (Zahle et al., 2024). Detailed and verified models for these turbines are included with the WISDEM code.

Table 3 breaks down the total CAPEX, blade cost, generator cost, and tower cost for the four reference turbines, as well as

the percent contribution to total CAPEX. Results from WISDEM indicate that the largest contributor to modern off-shore wind turbine CAPEX is now the generator at approximately 40%, while the remaining components each contribute 20%.



**Table 3.** CAPEX breakdown by component, including component percentage contribution to CAPEX, for three reference wind turbines as estimated using WISDEM.

|  | DTU 10-MW | IEA 10-MW | IEA 15-MW | IEA 22-MW | Average |
|---|---|---|---|---|---|
| CAPEX [$1000] | 13,686.57 | 13,973.24 | 15,497.56 | 30,386.34 |  |
| Blade Cost [$1000] | 2,329.57 | 2,322.91 | 3,182.22 | 5,429.85 |  |
| % CAPEX | 17% | 17% | 23% | 18% | 20% |
| Generator Cost [$1000] | 5,158.65 | 5,702.80 | 5,003.33 | 11,810.72 |  |
| % CAPEX | 38% | 42% | 37% | 39% | 40% |
| Tower Cost [$1000] | 3,630.95 | 3,325.48 | 2,459.91 | 4,105.06 |  |
| % CAPEX | 26% | 24% | 18% | 14% | 20% |
| % Remainder | 19% | 17% | 22% | 29% | 20% |

These results for large offshore wind turbines with modern technologies indicate that the balance of capital costs have changed from the original formulation in Eq. (11), demonstrating that increased generator sizes and changing generator materials are increasing generator cost. This leads to the following updated relationship for $\Delta CAPEX$:

$$\Delta CAPEX = 0.2\left(\Delta M_{rotor}\right) + 0.2\left(\Delta S_{tower}\right) + 0.4\left(\Delta Q_{shaft}\right) + 0.2 \tag{12}$$

A comparison of the $\Delta CAPEX$ model coefficients from Buck and Garvey (2015) and those derived here in the present work is also shown in Table 4.

**Table 4.** Coefficients of proportionality for $\Delta CAPEX(a)$.

| Reference | A | B | C | D |
|---|---|---|---|---|
| Buck and Garvey (2015) | 0.2 | 0.2 | 0.2 | 0.4 |
| Present Work | 0.2 | 0.2 | 0.4 | 0.2 |

This section derived a relationship for $\Delta CAPEX$ as a function of design axial induction factor. Based on these relationships, the trend of $\Delta CAPEX$ versus design axial induction factor requires additional investigation. This will be accomplished later

in Section 4 for a given reference turbine configuration as the baseline. In addition, new relationships were proposed for rotor mass and relative distribution of CAPEX costs that are representative of modern off-shore wind turbine sizes and technologies. The sensitivity of $\Delta CAPEX$ to these new relationships will also be discussed in more detail later in Section 4 with the goal of fully understanding the CAPEX trends for up-scaled rotors.





Combining the relationships for the components of $\Delta CAPEX$ as a function of blade radius, $R(a)$, derived in the proceeding sections, we arrive at the final form of $\Delta CAPEX(a)$

$$\Delta CAPEX(a) = A\left(\frac{R(a)}{R^0}\right)^X + B\left[\frac{4a(1-a)}{C_T^0}\left(\frac{R(a)}{R^0}\right)^2\right] + C\left(\frac{R(a)}{R^0}\right) + D$$

$$\text{subject to} \quad \longrightarrow \quad \frac{R(a)}{R^0} = \left(\frac{C_T^0}{4a(1-a)}\right)^{1/3}$$

(13)

where the values of $X$ for the exponent of the $\Delta M_{rotor}(a)$ term are given in Table 2 and the $\Delta CAPEX$ coefficients of proportionality for the four major components are given in Table 4.

## 3 Operations & Maintenance Cost (OPEX) Model

With higher failure rates and increased cost of repair and maintenance, operations and maintenance (OPEX) costs for offshore wind turbines are nearly twice that of their onshore equivalents and can contribute to as much as 30% of the total LCOE (Dao et al., 2020; Dinwoodie et al., 2013; Stehly et al., 2020). Despite this significant contribution to LCOE, OPEX is generally difficult to predict as there is little field data available to the public to better inform OPEX modeling (Dao et al., 2020). Additionally, as blade length increases for an up-scaled rotor, rotor loads change which can have a significant effect on OPEX planning and costs. To better model OPEX and understand how OPEX changes with rotor loading, the present work also seeks to develop a simple equation for $\Delta OPEX$ costs as a function of design axial induction factor, $a$. A few recent works (Carroll et al., 2016; Dao et al., 2019, 2020) have analyzed what little field data are available and serve as guidance for the development of a $\Delta OPEX$ equation.

Adapting a relationship proposed by Dao et al. (2019) for the basis of the present work, OPEX is approximated as a sum of fixed ($OPEX_{fixed}$) and variable ($OPEX_{variable}$) costs, and the variable OPEX component is additionally split into a planned maintenance ($C_{PM}$) and cost of random failures ($C_{CM}$) component:

$$OPEX = OPEX_{fixed} + OPEX_{variable} = OPEX_{fixed} + C_{PM} + C_{CM}$$

(14)

To estimate the change in OPEX ($\Delta OPEX$) costs between a reference rotor and a new up-scaled rotor, each term of Eq. (14) can be divided by a reference value to yield the following approximation:

$$\Delta OPEX = E * \left(\frac{OPEX_{fixed}}{OPEX_{fixed}^0}\right) + F * \left(\frac{C_{PM}}{C_{PM}^0}\right) + G * \left(\frac{C_{CM}}{C_{CM}^0}\right)$$

(15)

Here $E$, $F$, and $G$ are again proportionality constants akin to those in the $\Delta CAPEX$ function and will be derived later using available OPEX data.





According to Dao et al. (2019), $OPEX_{fixed}$ is not a function of wind turbine component failure rate or downtime, but rather are assumed to be a flat rate of 10% of CAPEX such that $OPEX_{fixed} = 0.1 * CAPEX$. Normalizing this by the reference
value yields the following expression for $\Delta OPEX_{fixed}$

$$\Delta OPEX_{fixed} = \frac{OPEX_{fixed}}{OPEX_{fixed}^0} = \Delta CAPEX \tag{16}$$

where $\Delta CAPEX$ is the same relationship derived and discussed in detail in Section 2. The two variable costs are, however, a function of total failure rate, $\lambda$, defined as the total number of failures per year across all subsystems. Dao et al. (2019) propose the following relationship for the planned maintenance costs, $C_{PM}$,

$$C_{PM}(a) = C_{PM}^0 \left( \frac{\lambda^0}{\lambda(a)} \right) \tag{17}$$

where $\lambda^0$ is the baseline wind turbine failure rate, $C_{PM}^0$ is the baseline cost of the planned maintenance strategy, and $\lambda(a)$ is the new failure rate. It should be noted that this relationship tells us that as failure rate increases from the baseline, $C_{PM}$ decreases. Though this relationship seems counter-intuitive, the trend tells us that the more cost contributed to planned maintenance activities, the more likely a failure-inducing problem is to be caught earlier and the less-likely an unplanned failure is to occur
(Dao et al., 2019).

If we assume that $\lambda$ represents all of the failures from all subsystems, then $\lambda = \sum_{i=1}^{N} \lambda_i$ for all $i$ subsystems. Dividing by the reference value of $C_{PM}^0$ and using the definition of $\lambda$, we can express the change in unplanned maintenance between a reference turbine and a new turbine as

$$\Delta C_{PM}(a) = \frac{C_{PM}}{C_{PM}^0} = \sum_{i=1}^{N} \left( \gamma_i \frac{\lambda_i^0}{\lambda_i(a)} \right) \tag{18}$$

where $\gamma_i$ ($\sum \gamma_i = 1$) is a proportionality constant introduced to the $\Delta C_{PM}$ term to account for the fact that some subsystems fail more frequently than others.

Dao et al. (2019) also provide a similar relationship for unplanned maintenance activities, $C_{CM} = \sum \lambda_i \times CoF(d_i)$. In this equation, $CoF(d_i)$ is the cost of failure and is given by $CoF(d_i) = c_r + p_r \times d_i \times c_{labor}$, where $d_i$ is wind turbine downtime required to repair the $i^{th}$ component, $c_r$ is cost of repair, $p_r$ is proportion of downtime used for repairs ($0 \le p_r \le 1$), and $c_{labor}$
is cost of labor. Dividing by the reference value and assuming that downtime, cost of repair, and repair time proportion remain the same between the reference and up-scaled turbine for all subsystems, we derive the following approximation for changes in unplanned maintenance:

$$\Delta C_{CM} = \sum_{i=1}^{N} \left( \kappa_i \frac{\lambda_i(a)}{\lambda_i^0} \right) \tag{19}$$





Here, $\kappa_i$ ($\sum \kappa_i = 1$) is a proportionality constant introduced to the change in the unplanned maintenance term to account for
the fact that $CoF(d_i)$ varies by subsystem. This results in the final generic form of the proposed $\Delta OPEX$ function:

$$\Delta OPEX(a) = E * (\Delta CAPEX(a)) + F * \left( \sum_{i=1}^{N} \gamma_i \frac{\lambda_i^0}{\lambda_i(a)} \right) + G * \left( \sum_{i=1}^{N} \kappa_i \frac{\lambda_i(a)}{\lambda_i^0} \right) \tag{20}$$

Note that though $OPEX_{fixed} = 0.1 * CAPEX$ and the unplanned maintenance costs, $C_{PM}^0$ and $CoF(d_i)$, are eliminated
in the $\Delta OPEX$ equation, these values are still needed to determine the proportional contribution of $\Delta OPEX_{fixed}$, $\Delta C_{PM}$,
and $\Delta C_{CM}$ to total $\Delta OPEX$, i.e. the coefficients $E$, $F$ and $G$, respectively. This process is discussed in more detail in Section
310 3.2.

### 3.1 Wind Turbine Subsytems

Wind turbines are complex machines composed of many highly integrated subsystems in order to transform wind into usable
electricity. These subsystems include, but are not limited to, the following: the rotor, drivetrain, gearbox (for geared turbines),
generator, hydraulic system, yaw system, control system, the electrical, sensors, nacelle, and the tower/foundation structure.
With as many subsystems come just as many modes of failure, but they can be categorized into three main types: mechanical
(fracture, rupture, thermal, etc.), electrical (insulation, software fault, electrical failure, etc.), and material (fatigue, structural)
(Arabian-Hoseynabadi et al., 2010). With the goal of the present work being to investigate an aerodynamic optimum that
balances blade loads, power production, and cost, only subsystems prone to fatigue and mechanical overload failures due to
turbine blade loads are considered here in the OPEX function.
Fatigue of wind turbine components is due to fluctuations in loads resulting from atmospheric gusts with variable amplitude
fluctuations in wind speed at a range of frequencies. Subsystems with the highest contributions to OPEX that are susceptible
to fatigue and overloading failures as a result of increasing blade lengh for up-scaled turbines are the blade root sections, the
pitch mechanism, the drivetrain, and the generator. Note that the drivetrain and generator are subsystems that are composed
of multiple components. The drivetrain can include rotor bearings, low-speed shaft, and gearboxes (Nejad et al., 2022; Ozturk
et al., 2018). Though the generator is sometimes included in the definition of the drivetrain, here it is considered separately
as its own complex subsystem comprised of windings, generator brushes, and bearings that are responsible for converting the
torque load transmitted by the drivetrain from the rotor into power (Ozturk et al., 2018; Packer, 2015).
Not only are these four subsystems prone to failure modes from blade loads, these are the four that contribute the most
OPEX costs. From a review of several studies that compiled and analyzed wind turbine reliability data from operational wind
farms, the subsystems with the highest failure rates are the pitch system (11 - 20%), rotor blades (3 - 8%), generator (8 - 18%)
and drivetrain (5 - 8%) (Anderson et al., 2023; Carroll et al., 2016; Dao et al., 2019; Li et al., 2020; Tavner, 2012). Though the
blades and drivetrain seem to have relatively low failure rates compared to the pitch bearing and generator, they have some of
the highest contributions to turbine downtime for repair or maintenance (Dao et al., 2019; Carroll et al., 2016; Gómez et al.,
2021), the most significant contribution to turbine shutdowns as a result of a fault (Li et al., 2020) and highest repair cost
(Carroll et al., 2016; Dao et al., 2020).





With the turbine subsystems of interest identified that have significant contributions to failure rate and/or overall maintenance costs, the components of the $\Delta OPEX$ equation that are dependent on subsystem failure rates can be defined. First, the change in planned maintenance cost ($\Delta C_{PM}$) can be broken down into the relevant components to this optimization problem as follows:

$$\Delta C_{PM} = \gamma_B \left(\frac{\lambda_B^0}{\lambda_B}\right) + \gamma_P \left(\frac{\lambda_P^0}{\lambda_P}\right) + \gamma_D \left(\frac{\lambda_D^0}{\lambda_D}\right) + \gamma_G \left(\frac{\lambda_G^0}{\lambda_G}\right) + \gamma_O \tag{21}$$

Here, $\gamma_B$ is the blade, $\gamma_P$ the pitch bearing, $\gamma_D$ the drivetrain, and $\gamma_G$ the generator proportionality constants to indicate total contribution to $C_{PM}$, respectively. Any $C_{PM}$ failures not covered by blade, drivetrain, or generator are considered "Other" and are accounted for by the proportionality constant $\gamma_O$. These constants are discussed later in Section 3.2 and represent the percent of total failures per turbine per year from that subsystem.

Finally, the change in unplanned maintenance cost ($\Delta C_{CM}$) can be broken down into the individual subsystems of interest as follows:

$$\Delta C_{CM} = \kappa_B \left(\frac{\lambda_B}{\lambda_B^0}\right) + \kappa_P \left(\frac{\lambda_P}{\lambda_P^0}\right) + \kappa_D \left(\frac{\lambda_D}{\lambda_D^0}\right) + \kappa_G \left(\frac{\lambda_G}{\lambda_G^0}\right) + \kappa_O \tag{22}$$

In Eq. (22), $\kappa_B$ is the blade, $\kappa_P$ the pitch bearing, $\kappa_D$ the drivetrain, and $\kappa_G$ the generator proportionality constants to indicate total contribution to $C_{CM}$, respectively. Any $C_{CM}$ cost not covered by blade, drivetrain, or generator repair is considered
"Other" and is accounted for by the proportionality constant $\kappa_O$. These constants are also discussed below in Section 3.2 and represent the percent of total unplanned maintenance cost contributed from that subsystem.

### 3.2    Model Coefficients from OPEX Data

With the general form of $\Delta OPEX$ and the expanded $C_{PM}$ and $C_{CM}$ terms based on relevant turbine subsystems with failure rates that are sensitive to changes in turbine loading, the next step is to determine all of the unknown proportionality constants
that remain. To do so requires an analysis of published field data for turbine failure rate, cost of repairs, and turbine downtime for repair. As most operations and maintenance companies consider this data proprietary, it is difficult to obtain extensive databases from which these coefficients should be derived.

The present work, as a result, is based on a limited set of published data from Dao et al. (2020) and Carroll et al. (2016) that are taken from operational wind farms. Though other published datasets exist, these two publications are identified by the au-
thors as the most extensive and complete datasets for the task here. In the dataset from Dao et al. (2020), there are three different wind farms with three different failure rates: CIRCE (Centro de Investigacion de Recursos y Consumos Energéticos) which had the lowest failure rates, LWK (Landwirtschaftskammer) which had slightly higher failure rates, and Strath (University of Strathclyde) with the highest failure rates.

First, the proportionality constants in Eq. (21) are determined using published failure rate field data, see Appendix B for a
summary of the data. Using the published data, first the total failures, $\lambda$, are determined by summing up all failure rates across





all subsystems in the dataset. Then, the fractional contribution to the total failure rate by each component is found by dividing the total failures of the component $\lambda_i$ by the total failures $\lambda$. Results for the wind farms with various failure rates are included in Table 5.

**Table 5.** Variation in proportionality coefficients for the different turbine subsystems of the $C_{PM}$ equation with different reliability datasets.

| dataset | $\gamma_B$ | $\gamma_P$ | $\gamma_D$ | $\gamma_G$ | $\gamma_O$ |
|---------|-------|-------|-------|-------|-------|
| CIRCE   | 0.238 | 0.157 | 0.086 | 0.157 | 0.362 |
| LWK     | 0.250 | 0.113 | 0.039 | 0.181 | 0.417 |
| Strath  | 0.062 | 0.134 | 0.078 | 0.120 | 0.606 |

The proportionality constants in Eq. (22) are determined using a similar method. Whereas the $\gamma_i's$ only require failure rate
data, determining the proportional contribution to $C_{CM}$ costs requires cost of failure ($CoF$) data, including cost of repair, downtime, and cost of labor. Tables detailing these costs are also included in Appendix B for the four different cases. By following the formula for $C_{CM,i} = \sum[\lambda_i * (c_r + p_r \times d_i \times c_{labor})]$, the unplanned maintenance cost for each component is computed using the published data. The total $C_{CM}$ is then computed, and the fractional contribution to the total $C_{CM}$ by each component is found by dividing the component unplanned maintenance cost $C_{CM,i}$ by the total unplanned maintenance cost
$C_{CM}$. Results are included in Table 6.

**Table 6.** Variation in proportionality coefficients for the different turbine subsystems of the $C_{CM}$ equation with different reliability datasets.

| dataset | $\kappa_B$ | $\kappa_P$ | $\kappa_D$ | $\kappa_G$ | $\kappa_O$ |
|---------|-------|-------|-------|-------|-------|
| CIRCE   | 0.250 | 0.079 | 0.159 | 0.196 | 0.316 |
| LWK     | 0.244 | 0.037 | 0.201 | 0.117 | 0.402 |
| Strath  | 0.007 | 0.016 | 0.199 | 0.053 | 0.725 |

Finally, using the DTU 10-MW as the reference wind turbine and the $C_{CM}$ approximations from the Dao et al. (2019) and Carroll et al. (2016) datasets, the percent contribution of all three components of $\Delta OPEX$ in Eq. (15) are determined. First, CAPEX for the reference turbine is again estimated using WISDEM. Recall that Dao et al. (2019) assumed that $OPEX_{fixed} \sim$ 10% CAPEX, an assumption used in this model. Additionally, Dao et al. (2019) assumed that $C^0_{PM} \sim 15\%$ CAPEX. Using the
WISDEM estimate for CAPEX, the values for $OPEX_{fixed}$ and $C^0_{PM}$ are known. Combined with the total $C_{CM}$ calculated for the different datasets to determine the $\Delta C_{CM}$ coefficients, fractional contribution of each term to the total OPEX can be calculated. The table of the $OPEX_{fixed}, C^0_{PM}$, and $C_{CM}$ data are included in Appendix B. The final proportional contributions are given here in Table 7.

### 3.3 Failure Rate via Damage-Equivalent Loads

With the equations determined for $\Delta OPEX$, the final step is to link the $\Delta OPEX$ model to the rotor design axial induction factor, $a$. To do so, it is desirable to investigate if a relationship exists between failure rate of the turbine subsystems of interest,



**Table 7.** Variation in proportionality coefficients for $\Delta OPEX$ with different reliability datasets.

| dataset | $E$ | $F$ | $G$ |
|---------|-----|-----|-----|
| CIRCE | 0.40 | 0.598 | 0.002 |
| LWK | 0.40 | 0.59 | 0.01 |
| Strath | 0.37 | 0.56 | 0.07 |

$\lambda_i$, and blade length, $R$, since $R$ was defined as a function of axial induction factor, $a$, via the low-induction objective function. Predicting component failure rate is difficult, however, for turbines in the preliminary design phase where the full structural definition is unknown and there are many complex factors that lead to component failure in the field. Additionally, while the academic reference turbine designs used frequently as baselines have published detailed structural layups, these structural designs are meant to serve as a reference to the community at large and may not result in representative failure behavior over the operational lifetime.

An alternative measure of failure rate considered in the present work is damage-equivalent load (DEL). The process of computing the damage-equivalent load involves converting the true load time series with an unsteady amplitude and frequency into an equivalent fixed-amplitude, constant frequency series that produces the same lifetime damage. By assuming the same loading frequency for all equivalent-load series, DEL standardizes the comparison between a reference case and a proposed preliminary turbine design, only requiring a comparison in the amplitude of the equivalent-load series. From the definition of DEL, one can show that DEL is proportional to failure rate, $\lambda$,(Hayman, 2012); thus, changes in DEL are considered here as representative of predicting changes in failure rate.

For the present study, DEL is computed with the open-source NREL python-based tool pCrunch (Nunemaker and Abbas, 2024), which takes an unsteady loading time series as an input. For the present work, the unsteady time series used to estimate fatigue DEL is representative of design load condition (DLC) 1.2. from the International Electrotechnical Commission (IEC) 61400-1 - the international standard for wind turbine design. DLC 1.2 is classified as normal power production ($V_{in} < V_{hub} < V_{out}$) assuming a Normal Turbulence Model (NTM) (IEC, 2005) and is the standard for fatigue estimates. The open-source tool TurbSim (Kelley, 2011) was used to generate the representative 10-minute time series for DLC 1.2, where NTM was simulated using the Kaimal turbulence model (Kaimal et al., 1972) for IEC class A characteristic turbulence levels. The TurbSim-generated DLC 1.2 wind time series was then used as an input to OpenFAST (open-source Fatigue, Aerodynamics, Structures, and Turbulence analysis code) to calculate the load time series for relevant turbine motions under DLC 1.2 wind conditions (Jonkman and Buhl, Jr., 2005).

To develop relationships for DEL versus blade radius, four reference wind turbines with verified aero-structural OpenFAST models are used to cover the range of radii representative of the various blade lengths for an offshore wind farm fleet. The NREL 5-MW (Jonkman et al., 2009) is chosen as it is representative of the current blade length for an operational offshore wind turbine. The IEA 10-MW (Bortolotti et al., 2019), IEA 15-MW (Gaertner et al., 2020), and 22-MW (Zahle et al., 2024) are considered here as representative of the larger-scale next-generation offshore turbines in development today.





From Section 3.1, the four subsystems of interest that are significantly prone to fatigue failure are the blades, pitch mechanism, drivetrain, and generator. The blades carry two main loads: the aerodynamic out-of-plane aerodynamic flap moment and the in-plane edgewise gravitational moment. Though they are also distributed along the blade and carried by the spar caps and shear webs, the highest concentration of these loads is seen at the blade root where it connects to the hub (Nijssen, 2022; Marín et al., 2009). The blade root is thus an area of particular interest for blade fatigue failures and the load approximation for this failure mode is the root-mean-square of the flap ($M_{Flap}$) and edgewise ($M_{Edge}$) moments:

$$DEL_B \propto M_{B_{rms}} = \sqrt{M_{Flap}^2 + M_{Edge}^2} \tag{23}$$

For the rotor pitch mechanism, failure of the pitch bearings are of particular interest to the present work. The pitch bearings are susceptible to fatigue, corrosion, and wear failure. Of these failure modes, rolling contact fatigue is one of the most common and is due to asymmetric rotor loading (Menck et al., 2020; Menck and Stammler, 2023). Pitch bearings are also prone to wear damage from smaller oscillations induced by torsional blade motions (Menck and Stammler, 2023). Adapting the definition of the pitch bearing fatigue equivalent load, $PB_{eq}$, discussed in Keller and Guo (2022), pitch bearing fatigue is approximated in the present work by the following combination of axial force, $F_a$, radial force, $F_r$, and the overturning moment, $M$:

$$DEL_P \propto PB_{eq} = 0.75F_{B_r} + F_{B_a} + \frac{2M_{B_{rms}}}{D_{pw}} \tag{24}$$

Here, the radial force is the root-mean-square of the two radial forces at the blade root, or $F_{B_r} = \sqrt{F_{B_x}^2 + F_{B_y}^2}$, the axial force is equivalent to $F_{B_z}$, the overturning moment is equivalent to $M_{B_{rms}}$, and $D_{pw}$ is the diameter of the pitch bearing ring.

As the subsystem directly connected to the spinning rotor and responsible for transmitting the mechanical loads to the generator, the main bearing and low-speed shaft of the drivetrain are directly impacted by changes in rotor loading. Gravitational load changes impact gear and bearing alignment and can contribute to wear and fatigue (Keller et al., 2018). Under variable rotor loading, the low-speed shaft is prone to two types of failure modes relevant to the present work: i) deformation failure from low-cycle fatigue due to wind turbulence or mechanical overload associated with sudden gusts and ii) fracture due to high-cycle fatigue at the 1 per rev blade passage frequency (Tavner, 2012). Similar to the equivalent load of the pitch bearing, an equivalent load for the main bearing/low-speed shaft system, $MB_{eq}$ is defined as:

$$DEL_D \propto MB_{eq} = 0.75F_{LSS_r} + F_{LSS_a} + \frac{2M_{LSS_{rms}}}{D_{LSS}} \tag{25}$$

Here, $F_{LSS_r}$ is the root-mean-square of the radial forces on the low-speed shaft and is equivalent to $\sqrt{F_{LSS_y}^2 + F_{LSS_z}^2}$. The axial low-speed shaft force, $F_{LSS_a}$, is equivalent to $F_{LSS_x}$, and the root-mean-square of the low-speed shaft moments, $M_{LSS_{rms}}$, is equivalent to $\sqrt{M_{LSS_y}^2 + M_{LSS_z}^2}$), while $D_{LSS}$ is the diameter of the low-speed shaft.

Though the generator is most prone to electrical failures from faulty electronics, rotor loading can contribute to mechanical and electrical failures. One of the highest contributors to this is generator bearing failure (Olabi et al., 2021). Generator bearings



may fail due to high-cycle fatigue from repetitive loading and unloading of the generator (Packer, 2015; Pulikollu et al., 2023).

Asymmetric rotor loading transferred to the generator through the drivetrain can also cause rotor bar cracking failure, which is one of the most expensive types of generator failure modes (Olabi et al., 2021). Rotor speed fluctuations and mechanical torque overloading of the generator due to momentary torque peaks from wind gusts can also contribute to generator failures (Packer, 2015; Pulikollu et al., 2023). Since the rotor torque is the primary load handled by the generator and fluctuations from gusts can contribute to generator failure, the failure rate of the generator is approximated by changes in the aerodynamic rotor

torque, $Q_{Aero}$:

$$DEL_G \propto Q_{Aero} \tag{26}$$

A graphical representation of the relevant loads that comprise $M_{B_{rms}}$, $PB_{eq}$, $Q_{Aero}$, and $MB_{eq}$ is included below in Figure 4.

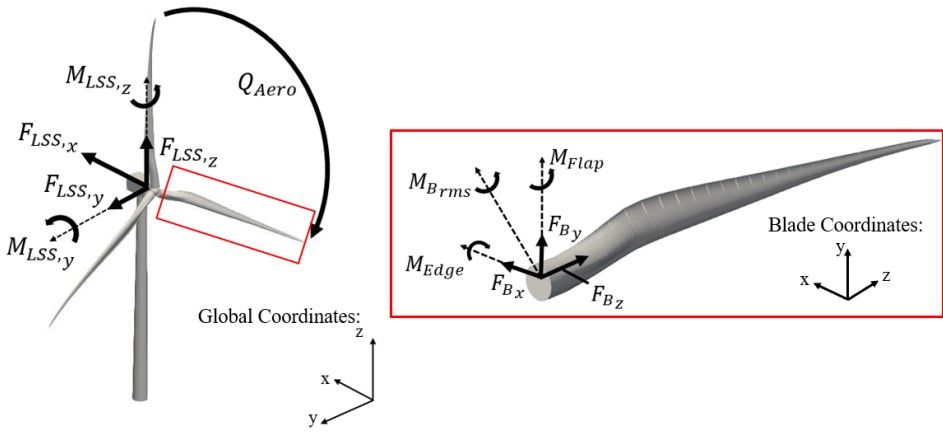

**Figure 4.** Definition of the relevant moments and forces comprising the blade root ($M_{B_{rms}}$), pitch bearing ($PB_{eq}$), main bearing ($MB_{eq}$), and rotor torque ($Q_{Aero}$) moments.

Ten-minute time series for the blade root forces ($F_{B_x}$, $F_{B_y}$, $F_{B_z}$), blade root moments ($M_{Flap}$, $M_{Edge}$), LSS forces ($F_{LSS_x}$, $F_{LSS_y}$, $F_{LSS_z}$), LSS moments ($M_{LSS_y}$, $M_{LSS_z}$) and aerodynamic torque ($Q_{Aero}$) are collected in OpenFAST for the represen-

tative DLC 1.2 wind conditions, and DELs are computed using pCrunch. The resulting pCrunch DELs (symbols) are plotted versus blade radius for all four loads of interest in Figure 5 and trend lines are plotted for each load case as an approximation for the DEL versus $R$ relationship.

The trend lines plotted in Figure 5 are also listed in Table 8, including R-Squared values to show goodness-of-fit for these approximations. For the four reference turbines used here and the four loads with notable contribution to OPEX that are

sensitive to changes in blade size and loading, it appears that DEL generally increases with $R^X$, similar to rotor mass. The trend of DEL as a function of rotor diameter being a power law was also found by Kazacoks and Jamieson (2015).



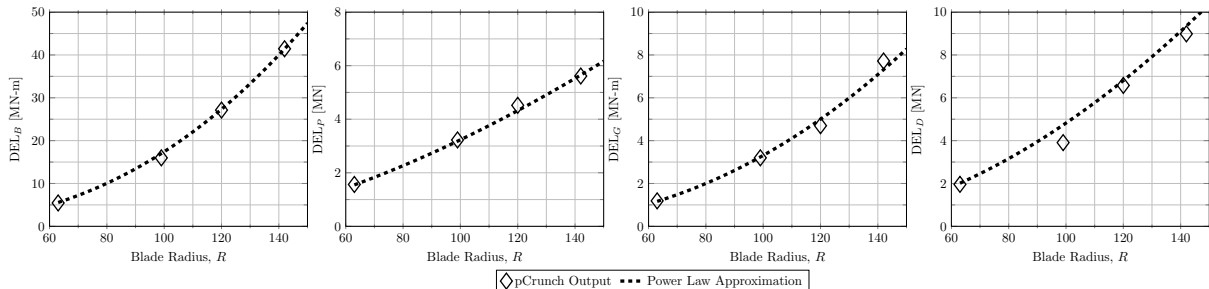

**Figure 5.** Estimated DEL trend for the blades (left), pitch (left-center), generator (right-center), and drivetrain (right) for four reference turbines.

**Table 8.** Power law approximations for DEL versus blade radius.

| Component | Load | Equation | R-Squared |
|---|---|---|---|
| Blades | $DEL_B$ | $0.0002 * R^{2.50}$ | 0.9993 |
| Pitch | $DEL_P$ | $0.0021 * R^{1.59}$ | 0.9966 |
| Generator | $DEL_G$ | $0.0001 * R^{2.26}$ | 0.9908 |
| Drivetrain | $DEL_D$ | $0.0008 * R^{1.89}$ | 0.9888 |

Note that approximations for these four DELs versus blade radius are limited to available data. The suite of fully-defined reference offshore wind turbine geometries is limited. As more reference turbine geometries with detailed structural informa­tion become available, these relationships should be re-investigated with additional data over the wide range of blade radii

representative of large offshore wind turbines.

Combining the relationships for $\Delta OPEX$ and $DEL$ as a function of blade radius, $R(a)$, derived in the proceeding sections, we arrive at the final form of $\Delta OPEX(a)$

$$
\begin{aligned}
\Delta OPEX = E * &\left[ A \left( \frac{R(a)}{R^0} \right)^X + B \left[ \frac{4a(1-a)}{C_T^0} \left( \frac{R(a)}{R^0} \right)^2 \right] + C \left( \frac{R(a)}{R^0} \right) + D \right] \\
+ F * &\left[ \gamma_B \left( \frac{R^0}{R(a)} \right)^{2.50} + \gamma_P \left( \frac{R^0}{R(a)} \right)^{1.59} + \gamma_D \left( \frac{R^0}{R(a)} \right)^{2.26} + \gamma_G \left( \frac{R^0}{R(a)} \right)^{1.89} + \gamma_O \right] \\
+ G * &\left[ \kappa_B \left( \frac{R(a)}{R^0} \right)^{2.50} + \kappa_P \left( \frac{R(a)}{R^0} \right)^{1.59} + \kappa_D \left( \frac{R(a)}{R^0} \right)^{2.26} + \kappa_G \left( \frac{R(a)}{R^0} \right)^{1.89} + \kappa_O \right] \\
\text{subject to} \quad \longrightarrow \quad &\frac{R(a)}{R^0} = \left( \frac{C_T^0}{4a(1-a)} \right)^{1/3}
\end{aligned}
\tag{27}
$$

where the $\gamma_i$'s are given in Table 5, the $\kappa_i$'s in Table 6, and the coefficients $E$, $F$, and $G$ in Table 7.



# 4   The Fatigue, Aerodynamics, and Cost-scaled Turbine (FACT) Objective Function

The final component of LCOE is the annual energy production (AEP), which can have a sizeable impact on compensating for increased turbine costs to keep cost-of-energy low. A special function of the low-induction rotor design function used in this paper to represent blade radius as a function of $a$ is the relationship between axial induction and annual energy production (AEP). Annual energy production, assuming there are 8760 hours per year, is defined as $AEP = 8760 * \overline{P}$, where $\overline{P}$ is the average wind turbine power produced at freestream wind speed $V_o$. The average power for a discrete wind speed distribution is given by

$$\overline{P} = \sum_V P(V)p(V) \tag{28}$$

where $P(V)$ is the power produced for a given wind speed $V$ and $p(V)$ is the probability density function of wind speeds $V$ at a given wind site given by the general Weibull distribution, whose shape is determined by the scale factor $k$ and shape factor $c$ that are properties of the wind site where the turbine will be placed. For the present work, it is assumed that $k = 2$ such that probability density function takes on a special form known as the Rayleigh distribution:

$$p(V) = \frac{\pi}{2} \left( \frac{V}{\overline{V}^2} \right) exp \left[ -\frac{\pi}{4} \left( \frac{V}{\overline{V}} \right)^2 \right] \tag{29}$$

In Eq. (29), $\overline{V}$ is the average wind speed at the site, which is assumed to be 8 m/s for all case studies investigated here. To define AEP as a function of axial induction factor for the new up-scaled rotor designs, $P(V)$ can be defined as a function of axial induction factor, $P(a, V)$, from momentum theory as

$$P(V) = P(a, V) = 2\rho V^3 a(1-a)^2 \pi R(a)^2 \tag{30}$$

Note that this approximation over-estimates power at higher wind speeds. To account for the over-estimation, once the $P(a, V)$ exceeds $P_{rated}$, the routine sets $P(a, V) = P_{rated}$. To develop the final expression for $\Delta AEP$, AEP is normalized by the reference value as follows:

$$\Delta AEP(a) = \frac{\sum_V P(a, V) * p(V)}{\sum_V P^0(V) * p(V)} = \frac{\sum_V 4a(1-a)^2 V^3 R(a)^2 p(V)}{\sum_V 4a^0(1-a^0)^2 V^3 (R^0)^2 p(V)} \tag{31}$$

Here, $P^0(V) = 0.5\rho 4a^0(1-a^0)^2 V^3 \pi (R^0)^2$ is the power curve of the reference turbine. Though the power curves are both multiplied by the Rayleigh distribution, $p(V)$, which is assumed to not change between reference and up-scaled turbine, $p(V)$ cannot be eliminated from Eq. (31) as it is not constant over the summations. With the lower axial induction factor and higher blade radius, the low-induction up-scale rotor sees an increase in Region II power (Chaviaropoulos et al., 2013; Major et al.,



2022; Schmitz, 2020), thus the term $P(V) * p(V)$ changes slightly with design axial induction factor, $a$, so the Rayleigh distribution must remain within the $\Delta AEP$ equation.

Coupling the effect of the low-induction function for blade radius, which has been shown to reduce fatigue of major turbine motions and increase AEP (Major et al., 2022), to the LCOE equation and developing approximations for CAPEX and OPEX as functions of axial induction factor leads to the combined Fatigue, Aerodynamics, and Cost-scaled Turbine (FACT) objective function:

$$\Delta LCOE(a) = \frac{\overbrace{\Delta CAPEX(a)}^{Eq.(13)} + \overbrace{\Delta OPEX(a)}^{Eq.(20)}}{\underbrace{\Delta AEP(a)}_{Eq.(31)}} \tag{32}$$

The case studies that follow in this section are to determine the optimum range of design axial induction factors for lowest LCOE. Recall that the design axial induction factor is not necessarily the operating axial induction factor for the blade nor the local, spanwise axial induction factor. The purpose of identifying an ideal design axial induction factor is that it can be used to find an optimum spanwise circulation distribution from which the rotor planform can be designed using inverse design methods (Major et al., 2022; Schmitz, 2020). For the following case studies, five different reference wind turbines are used as the baseline from which an up-scaled turbine is designed: the DTU 10-MW Reference Wind Turbine (Bak et al., 2013), the IEA 10-MW Reference Wind turbine (Bortolotti et al., 2019), the IEA 15-MW Reference Wind Turbine (Gaertner et al., 2020), a re-designed version of INNWIND 20-MW Wind Turbine (Sartori et al., 2018), and the IEA 22-MW Reference Wind Turbine (Zahle et al., 2024). These turbines cover a wide range of rated powers for the current and next-generation of offshore wind turbines, as well as different design methodologies. Details of each turbine, including the reference values needed for the various components of the $\Delta LCOE$ equation, are given in Table 9.

### 4.1 Sensitivity Study

The first study is dedicated to investigating the sensitivity of the optimum design axial induction factor for lowest LCOE to the various relationships derived in Sections 2 and 3 and to understand the general trends in $\Delta CAPEX$, $\Delta OPEX$, and $\Delta AEP$ versus design axial induction factor. In this case study, only the DTU 10-MW Reference Wind Turbine is used as the baseline turbine. This turbine is selected as the reference for the sensitivity analysis as it was designed to serve as a baseline from which the benefits of new technologies could be quantified (Bortolotti et al., 2019). Once the sensitivity of optimum axial induction factor is understood using a single baseline rotor, the sensitivity of the full $\Delta LCOE$ function will be investigated for all reference turbines listed in Table 9.

#### 4.1.1 CAPEX Analysis

In Section 2, relationships were defined for $M_{rotor}$, $S_{tower}$, and $Q_{shaft}$ in the $\Delta CAPEX$ equation as a function of axial induction factor. Additionally, $M_{rotor}$ was given for three different technology relationships, and two different sets of proportionality





**Table 9.** Reference turbines used in case study to determine optimum $a$-range for an up-scaled turbine. **Note that design axial induction factor is determined from Design $C_T$ using the momentum theory approximation that $C_T = 4a(1-a)$. It is not necessarily the operational axial induction factor.

| Parameter | DTU 10-MW RWT | IEA 10-MW RWT | IEA 15-MW RWT | INNWIND 20-MW | IEA 22-MW RWT |
|---|---|---|---|---|---|
| Wind Regime | IEC Class 1A | IEC Class 1A | IEC Class 1B | IEC Class 1C | IEC Class 1B |
| Rotor Orientation | Upwind | Upwind | Upwind | Upwind | Upwind |
| Rated Power, $P^0_{rated}$ | 10.0 MW | 10.0 MW | 15.0 MW | 20.0 MW | 22.0 MW |
| Blade Radius, $R^0$ | 89.0 m | 99.0 m | 120.0 m | 126.0 m | 142.0 m |
| Tower height, $H^0$ | 119 m | 119 m | 150 m | 163 m | 170 m |
| Tower width, $W^0$ | 9 m | 9 m | 10 m | 12 m | 10 m |
| Design Wind Speed, $V^0_o$ | 11.0 m/s | 10.5 m/s | 11.0 m/s | 11.4 m/s | 10.0 m/s |
| $TSR^0$ | 7.50 | 10.58 | 9.00 | 7.86 | 9.153 |
| Design $C^0_P$ | 0.476 | 0.464 | 0.489 | 0.442 | 0.500 |
| Design $C^0_T$ | 0.856 | 0.860 | 0.799 | 0.524 | 0.800 |
| Design $a^0$** | 0.330 | 0.280 | 0.310 | 0.170 | 0.310 |

constants for the contributions of each term of the $\Delta CAPEX$ equation were given. This section looks at the general trend

in $\Delta M_{rotor}$, $\Delta S_{tower}$ and $\Delta Q_{shaft}$, and investigates the variation in $\Delta CAPEX$ with the various rotor mass function and proportionality coefficients.

Trends of the $\Delta CAPEX$ components versus design axial induction factor are plotted in Figure 6 (left) for a proposed up-scaled version of the DTU 10-MW RWT. Note that it is assumed that the rated power and design wind speed of the up-scaled rotor is equivalent to that of the reference case. The variation in the $M_{rotor}$ term is also included for the three different rotor

mass relationships in Table 2.

As expected, $M_{rotor}$ increases notably with decreasing design axial induction factor. This is a direct result from the coupling to the low-induction objection function in Eq. (4), in which blade radius increases for decreasing axial induction factor to satisfy the bending moment constraint. Since the up-scaled rotor will always have a longer blade than the baseline rotor as a result of the low-induction objective function, it follows that the minimum $M_{rotor}$ is at the axial induction factor of the baseline

rotor. Comparing the three different approximations for $M_{rotor}$, the Fingersh et al. relationship predicts the highest increase in $M_{rotor}$, while the Reference + Commercial Turbine rotor mass estimate predicts the smallest change in $M_{rotor}$. For the longest blades (i.e., the lowest design axial induction factor), blade mass increase varies anywhere from 40% to almost 70%.

Similar to $M_{rotor}$, shaft torque, $Q_{shaft}$, increases for decreasing design axial induction factor. Recall the definition of $\Delta Q_{shaft}$ from Eq. (9). Since $\Delta Q_{shaft}$ goes with $R/R^0$, as blade radius increases with axial induction factor, it is expected that

shaft torque requirement will also increase. As a result of the coupling to the low-induction objective function, the tower structural moment, $S_{tower}$, is the only term that decreases with decreasing design axial induction factor. Applying the definition of $R/R^0$ in Eq. (4) to Eq. (7), this further reduces to $\Delta S_{tower} = [C_T(a)/C^0_T]^{2/3}$. Since $C_T(a) \leq C^0_T$ so that the bending moment



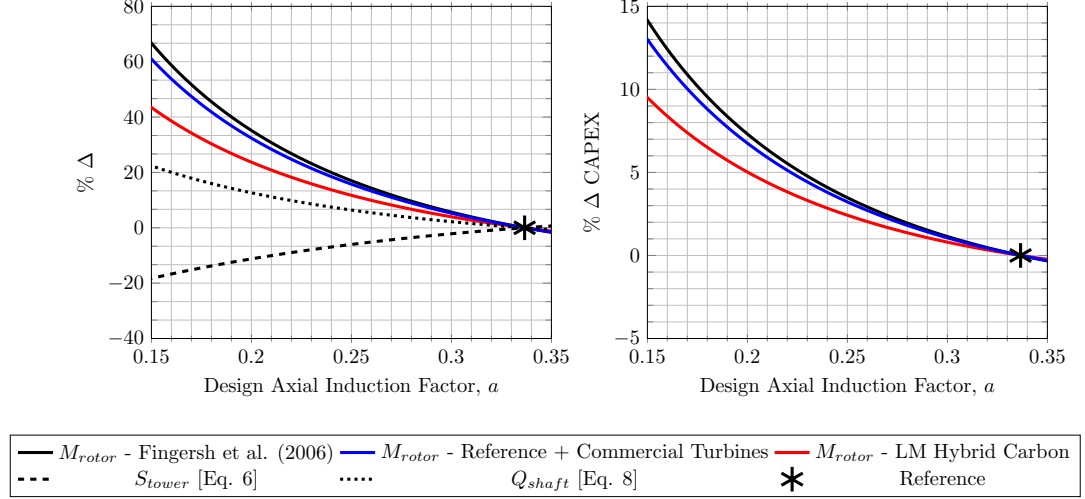

**Figure 6.** Percent change in rotor mass, tower thrust moment, and shaft torque as a function of design axial induction factor (left) and combined percent change in CAPEX for the three different rotor mass relationships (right); Reference rotor: DTU 10-MW RWT.

constraint is satisfied, the result is that the up-scaled rotor with longer blades operates at lower thrust, reducing the required tower structural capacity as axial induction factor decreases.

Using the baseline formulation of the $\Delta CAPEX$ equation in Eq. (11) where $M_{rotor}$, $S_{tower}$, and $Q_{shaft}$ have an equal 20% contribution to overall CAPEX, the general trend for $\Delta CAPEX$ versus design axial induction factor is plotted in Figure 6, right, for all three rotor mass relationships. In general, CAPEX is predicted to increase for decreasing design axial induction factor. Despite the reduced rotor thrust and tower structural cost, the notable increase in blade mass and required drivetrain torque increase the total CAPEX cost. This result indicates that to keep changes in CAPEX as low as possible requires the
smallest possible blades that can produce the desired rotor power.

Looking at the variation in $\Delta CAPEX$ with the rotor mass function, there is some notable variation between the three mass functions. While the predicted increase in rotor mass ranges from 40% to 70% (Figure 6, left), this only results in at most a 4% difference in $\Delta CAPEX$. The assumption that the rotor blades are only 20% of the total CAPEX means the significant variation in rotor mass predicted by the different $M_{rotor}$ functions is not necessarily the driving contributor to significant variations in
CAPEX.

Figure 7 investigates the sensitivity of the relative contribution of each component (rotor, tower, and drivetrain) to $\Delta CAPEX$ for the two different proportionality relationships presented in the present work: Eqs. (11) and (12). By using the $\Delta CAPEX$ relationship based on newer technologies (Eq. (12)), where the generator cost is the highest contributor to CAPEX at 40%, $\Delta CAPEX$ has at most a 14% increase for the lowest design axial induction factor investigated here. This is a modest rise in
potential CAPEX increase compared to the at most 9% increase for the original $\Delta CAPEX$ formulation published by Buck and Garvey (2015). As a more representative distribution of CAPEX costs for modern offshore wind turbines and a more



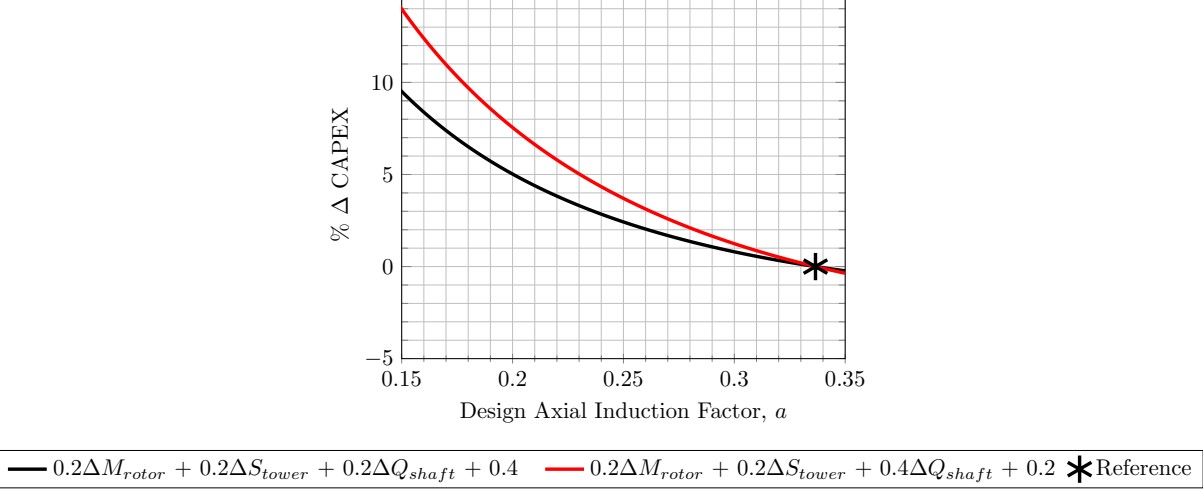

**Figure 7.** Percent change in CAPEX as a function of design axial induction factor for the two different sets of proportionality constants in $\Delta CAPEX$; Reference rotor: DTU 10-MW RWT.

conservative prediction with higher predicted CAPEX increase, Eq. (12) will be considered the representative relationship for all future analyses.

### 4.1.2 OPEX Analysis

In Section 3, relationships were defined for $OPEX_{Fixed}$, $C_{PM}$, and $C_{CM}$ in the $\Delta OPEX$ equation as a function of design axial induction factor. For $C_{PM}$, $C_{CM}$ and the overall $\Delta OPEX$ equation, three different sets of proportionality constants were given based on published reliability data for offshore wind farms. This section looks at the general trend of each contribution to $\Delta OPEX$ and investigates the variation of each term with the reliability database.

For the components of $\Delta OPEX$, Eq. (20), recall that it is assumed that $\Delta OPEX_{Fixed}$ is proportional to $\Delta CAPEX$, thus 570 the overall trend for $\Delta OPEX_{Fixed}$ is the same as $\Delta CAPEX$ as seen in Figures 6 (right) and 7. The two new relationships in $\Delta OPEX$ are for the planned and unplanned maintenance costs. Figure 8 plots $\Delta C_{PM}$ (left) and $\Delta C_{CM}$ (right) as a function of design axial induction factor. Each plot also contains three different lines representing the different sets of proportionality constants from Table 5 for $\Delta C_{PM}$ and Table 6 for $\Delta C_{CM}$ that correspond to the different contributions of each turbine component of interest to total unplanned cost for different reliability databases.

Looking at the change in planned maintenance cost, $\Delta C_{PM}$, in Figure 8 left, it is expected that $C_{PM}$ decreases with decreasing design axial induction factor. This is unique to the fact that $\Delta C_{PM}$ is proportional to $(1/\lambda)$, which implies that failure rate increases as planned maintenance activities decrease. Note that this result can be subject to different planned maintenance schedules to account for the increased turbine subsystem failure rates for turbines with longer blades.




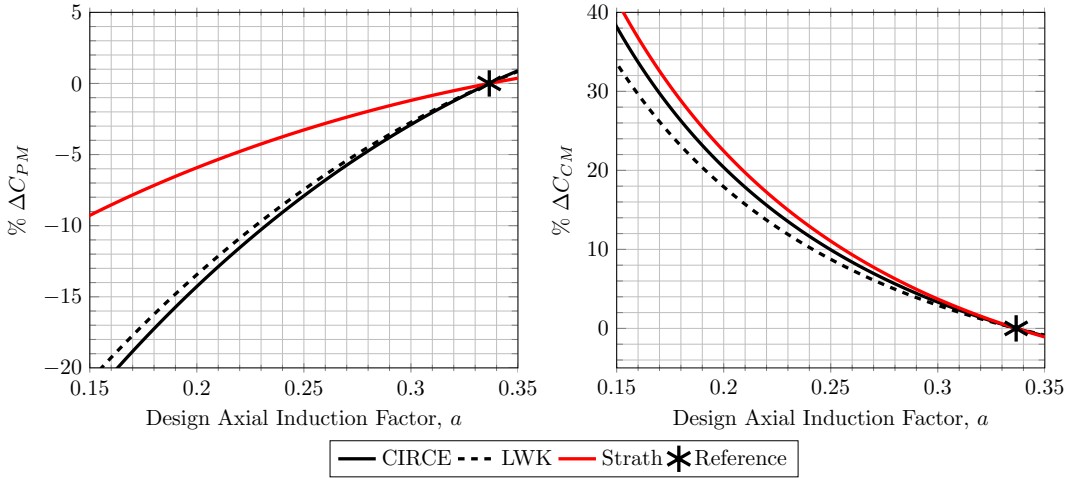

**Figure 8.** Percent change in $C_{PM}$ (left) and $C_{CM}$ (right) as a function of design axial induction factor for various failure rate databases; Reference rotor: DTU 10-MW RWT.

Comparing across datasets used to determine component contributions to total failure rate for $\Delta C_{PM}$, there is a notable
difference between $\Delta C_{PM}$ when using the failure rates from Strath (Carroll et al., 2016) and those from CIRCE and LWK (Dao et al., 2020). With the blades, pitch mechanism, drivetrain, and generator comprising less than 40% of the total failures, the Strath data leads to the smallest predicted reduction in $C_{PM}$ of at most 9%. Conversely, because the relevant subsystems investigated in the present work comprise more than 60% of the failures for both the CIRCE and LWK datasets, these reliability datasets result in a predicted reduction in $C_{PM}$ of more than 20%. Among the different reliability datasets from Dao et al.
(2020), the variation is much less notable than between the Dao et al. (2020) and Carroll et al. (2016) data.

In contrast to the $C_{PM}$ results, $\Delta C_{CM}$ increases notably with decreasing design axial induction factor. In this case, this term is proportional to $DEL \propto R^X$. As DEL increases for the blades, generator, drivetrain, and pitch mechanism with increasing blade length for lower design axial induction factors, unplanned maintenance, $C_{CM}$, increases with a power law. Unlike the results for $\Delta C_{PM}$, there is less variation in $\Delta C_{CM}$ with axial induction factor for the three different reliability datasets.
Unlike $\Delta C_{PM}$, the Carroll et al. (2016) dataset predicts higher $\Delta C_{CM}$ than the Dao et al. (2020) dataset. The relative cost contributions calculated for the Strath data for the four subsystems investigated here (see Table 6) are generally higher than those of the CIRCE and LWK datasets.

Combining these results, total $\Delta OPEX$ is plotted for the three different reliability datasets in Figure 9 left. Recall that the relative contributions for each term of $\Delta OPEX$ are listed in Table 7 for each reliability dataset. Near the baseline design point,
$a \approx 0.33$, there is no predicted change in $\Delta OPEX$, which is a consistent result. As design axial induction factor is decreased, OPEX initially decreases from the baseline value, reaches a minimum, and then increases exponentially for the smallest design axial induction factors considered here. The location of the minimum and predicted reduction in OPEX cost varies depending





on the reliability dataset. The Strathclyde reliability data appears to have little variation with design axial induction factor and predicts the smallest reduction in OPEX of only around 1% for $a \approx 0.18$. The CIRCE and LWK datasets predict much higher
reductions in OPEX of around 12% and 10%, respectively.

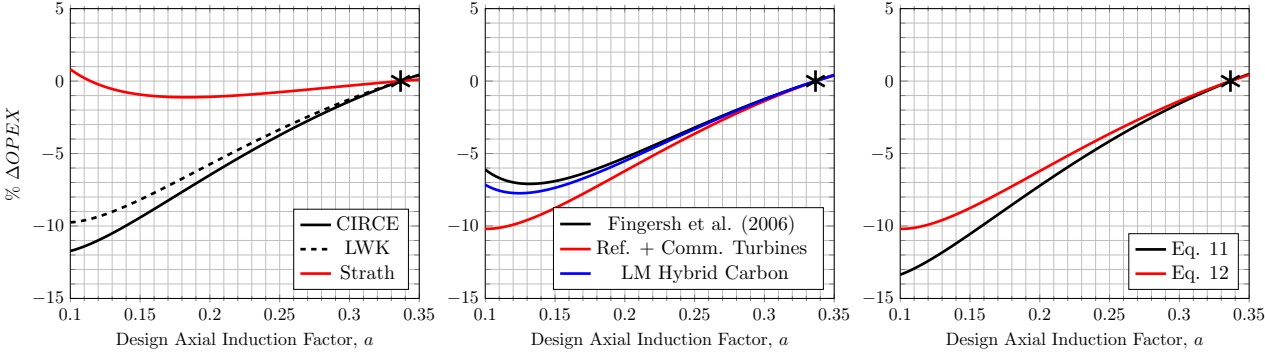

**Figure 9.** Effect of reliability database (left), rotor mass relationship (center), and CAPEX proportionality constants (right) on percent change in OPEX as a function of design axial induction factor; Reference rotor: DTU 10-MW RWT.

It is interesting to note that trends in $\Delta OPEX$ are notably different between the Carroll et al. (2016) and Dao et al. (2020) datasets, despite $\Delta C_{CM}$ having a higher increase than the possible reductions from $\Delta C_{PM}$. To investigate why, recall the $\Delta OPEX$ proportionality coefficients in Table 7. In general, $\Delta C_{PM}$ is about 59% of $\Delta OPEX$, while $\Delta C_{CM}$ is only 0.2% - 2%. Since $\Delta C_{PM}$ is the highest contributor to overall $\Delta OPEX$, the overall trend of $\Delta OPEX$ and variation with reliability
dataset mirrors that of $\Delta C_{PM}$. Because $C_{CM}$ is a relatively small fraction of total OPEX, $C_{CM}$ becomes a notable contribution to total OPEX at the smallest design axial induction factors where the percent change becomes significantly high.

Because $\Delta OPEX_{Fixed}$ is a function of $\Delta CAPEX$, and $\Delta CAPEX$ varies with both the rotor mass relationship and CAPEX proportionality constants, some variation in $\Delta OPEX$ is expected with the variation in $\Delta CAPEX$. In Figure 9 center, $\Delta OPEX$ is plotted for the three different rotor mass relationships that effect $\Delta CAPEX$, and in turn $\Delta OPEX_{Fixed}$. Similar
to $\Delta CAPEX$, $\Delta OPEX$ has a similar variation as rotor mass; however, the mass function that predicts the lowest increase in CAPEX (Ref. + Comm. Turbines) predicts the highest reduction in OPEX. A similar trend is observed for $\Delta OPEX$ by varying the form of the $\Delta CAPEX$ equation, as seen in Figure 9 right. The function that predicted the highest CAPEX increase (Eq. (12) with 40% generator contribution) predicts the highest reduction in OPEX.

### 4.1.3    Levelized-Cost of Energy

Minimizing CAPEX favors higher design axial induction factors near $a = 0.3$ where blade length is shortest, keeping rotor mass lighter and requiring smaller generators to meet torque requirements. From the results in Figure 9, minimizing OPEX in the total LCOE equation favors lower design axial inductions factors near $a = 0.1$ where $C_{PM}$ (the highest contributor to OPEX) has the highest reduction. This section now investigates the overall effect of these trends on $\Delta LCOE$ and the sensitivity of $\Delta LCOE$ to the rotor mass relationship, $\Delta CAPEX$ proportionality coefficients, and OPEX failure rate database.



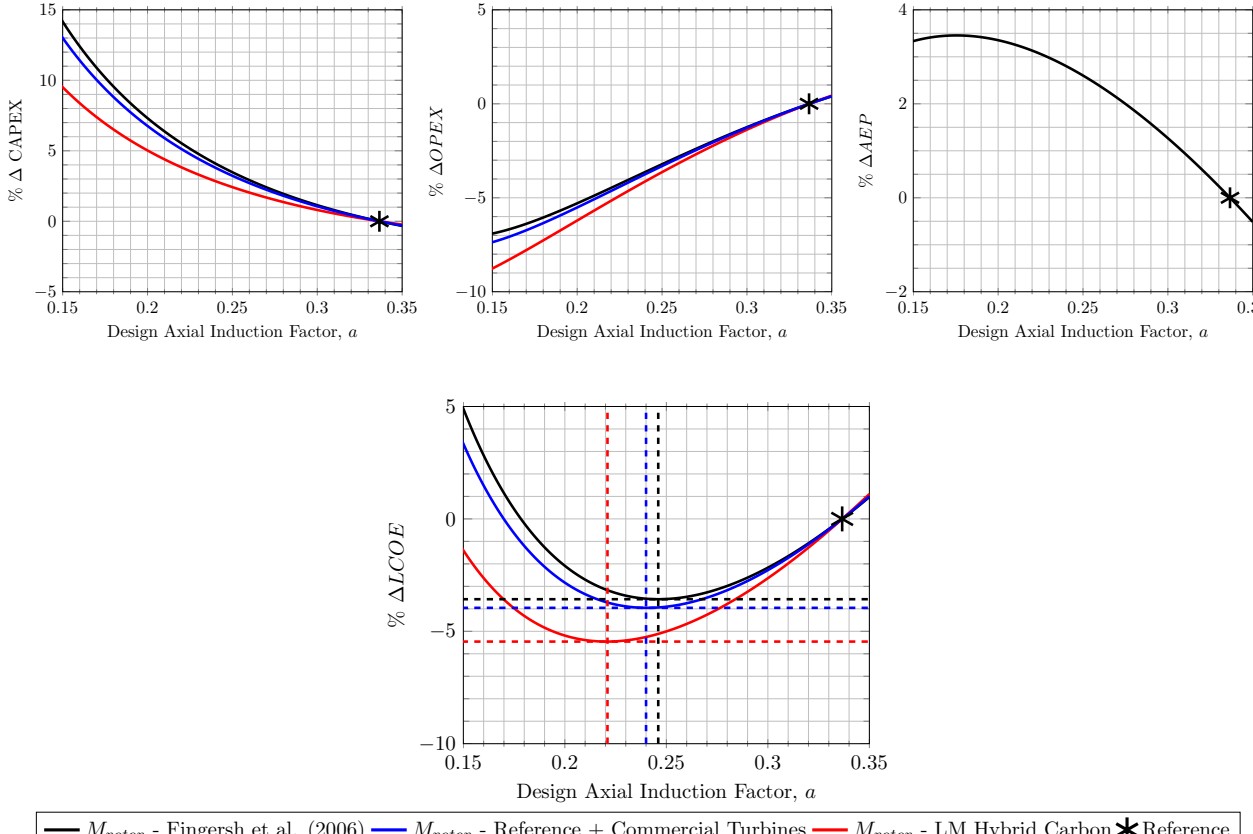

**Figure 10.** Sensitivity of percent change in CAPEX (top left), OPEX (top center), AEP (top right), and total LCOE (bottom) as a function of design axial induction factor to different rotor mass relationships; Reference rotor: DTU 10-MW RWT.

The variation in $\Delta LCOE$ with rotor mass relationship is plotted in Figure 10, bottom. Though the relationships for $\Delta CAPEX$ and $\Delta OPEX$ have been plotted in earlier figures, they are included at the top of Figure 10 for completeness to show the terms contributing to the overall $\Delta LCOE$. First, note the trend in $\Delta AEP$ for the DTU 10-MW in the top right of Figure 10. By increasing rotor blade length, the result of the low-induction objective function is a possible increase in AEP of the up-scaled rotor by as much as 3.5%. Though the up-scaled rotor is designed to produce the same rated power, the AEP increase comes from a notable increase in Region II power production (Major et al., 2022). Increasing AEP to reduce LCOE favors lower design axial induction factors, with the optimum for largest AEP increase being near $a = 0.18$ (i.e. the baseline low-induction rotor with a bending moment constraint). Note that actual improvements in AEP relative to the baseline rotor design are dependent on the Region II and III control strategy.

For the variation in overall $\Delta LCOE$ for each of the three rotor mass relationships seen in Figure 10 bottom, the result of the CAPEX, OPEX, and AEP trends is that $\Delta LCOE$ is minimized between $a = 0.22$ and 0.245 for the DTU 10-MW. By up-scaling the DTU 10-MW and designing a blade to operate in that axial induction factor range, this result indicates it is possible





to achieve a 3.5 - 5.5% reduction in LCOE. The highest possible LCOE reduction is observed with the updated Reference +
Commercial Turbine mass estimate.

Similarly, the trend for $\Delta LCOE$ for the two different $\Delta CAPEX$ equations is given in Figure 11. Compared to the effect
of the rotor mass relationships on LCOE, there is a notable difference in predicted LCOE reduction with choice of CAPEX
proportionality coefficients. The updated $\Delta CAPEX$ relation derived in the present work that favors increasing generator cost
gives the smallest predicted reduction in LCOE of 5.5% at a design axial induction factor of $a = 0.22$. The original $\Delta CAPEX$
relation taken from the work of Buck and Garvey (2015), predicts almost double the reduction in LCOE of nearly 9% for
design axial induction factor of $a = 0.19$. The new CAPEX function derived here, which emphasizes increasing generator costs

for large offshore wind turbines, is also a more conservative estimate for $\Delta LCOE$.

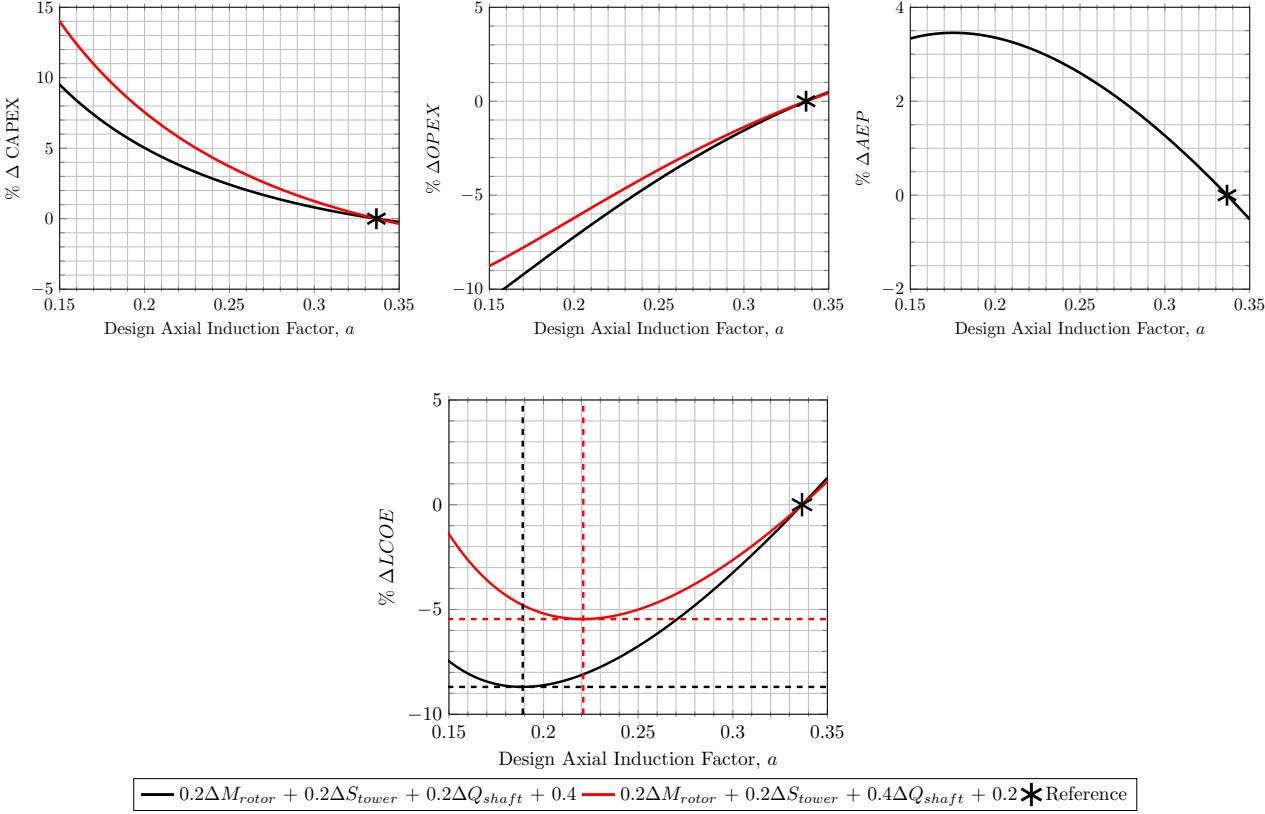

**Figure 11.** Sensitivity of percent change in CAPEX (top left), OPEX (top center), AEP (top right), and total LCOE (bottom) as a function of
design axial induction factor to different CAPEX proportionality constants; Reference rotor: DTU 10-MW RWT.

Finally, the effect of assumed failure rate database is investigated for overall $\Delta LCOE$ in Figure 12. Compared to Figures 10
and 11, the optimum range of design axial induction factor for an up-scaled FACT blade design does not appear to be any more
sensitive to the reliability database than compared to the variation due to the rotor mass or CAPEX function. For reliability
databases with different failure rates, downtime, and costs of repair, optimum design axial induction factor for minimum



$\Delta LCOE$ for an up-scaled rotor varies from $a = 0.227$ to $0.27$. Across this range of optimum design axial induction factors identified by the objective function, the predicted reduction in LCOE is somewhat sensitive to the assumed turbine reliability, ranging from -2% to -5%, with the Strath reliability data, with the highest assumed failure rate, giving the most conservative estimate. It is worth noting from Figure 12 bottom that higher assumed failure rates favor a higher design axial induction factor for a FACT rotor blade to minimize OPEX costs.

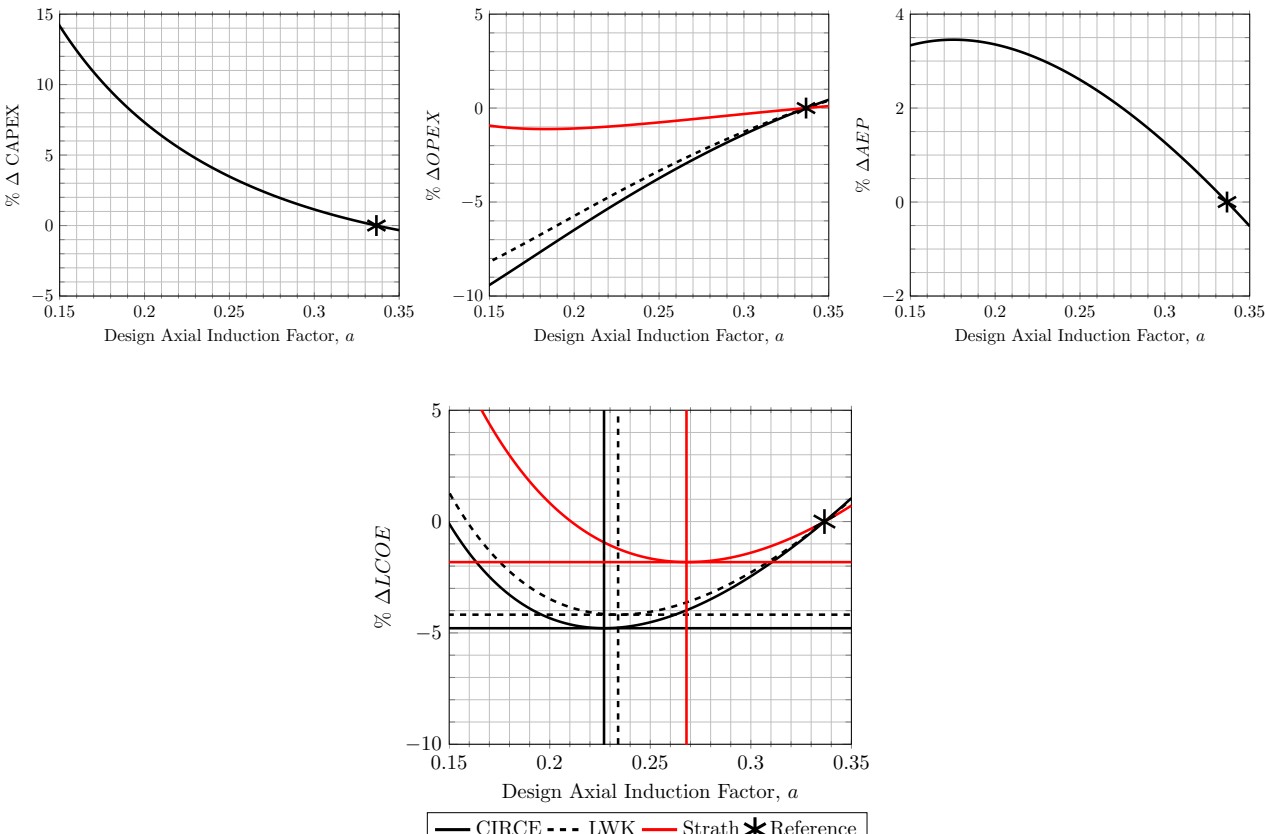

**Figure 12.** Sensitivity of percent change in CAPEX (top left), OPEX (top center), AEP (top right), and total LCOE (bottom) as a function of design axial induction factor to different failure rate databases; Reference rotor: DTU 10-MW RWT.

Across all the variations of $\Delta CAPEX$ and $\Delta OPEX$ investigated here, the optimum design axial induction factor for an up-scaled version of the DTU 10-MW RWT is between $a = 0.19 - 0.27$ Predicted reduction in LCOE ranges from 2% to 9%. With an overall range of 7% in possible LCOE reduction between the smallest and largest gains, the reduction that can be gained from the correct approximation of rotor mass, CAPEX cost distribution, and wind turbine reliability can be notable in terms of absolute LCOE.



## 4.2 Optimum Range of Design Axial Induction Factor for FACT Rotor

With the full trends of $\Delta CAPEX$, $\Delta OPEX$, $\Delta AEP$, and $\Delta LCOE$ understood from the case study with the DTU 10-MW in the previous section, this section applies those results to determine the optimum range of design axial induction factors at which to design a fatigue, aerodynamics, and cost-scaled turbine (FACT) rotor blade from the reference rotors listed in Table 9. Results are plotted in Figure 13. Solid lines in each plot are an estimate for LCOE reduction using old formulations of $\Delta CAPEX$ given in Buck and Garvey (2015), the original approximation for $M_{rotor}$ from Fingersh et al. (2006), and the conservative reliability data from Strath. Dashed lines in Figure 13 represent the $\Delta LCOE$ estimate based on the updated relationships developed in the present work. This estimate uses the Reference + Commercial turbine rotor mass relationship, the $\Delta CAPEX$ relationship that emphasizes increasing generator costs, and the $\Delta OPEX$ relationship based on the CIRCE reliability data with the lowest failure rates.

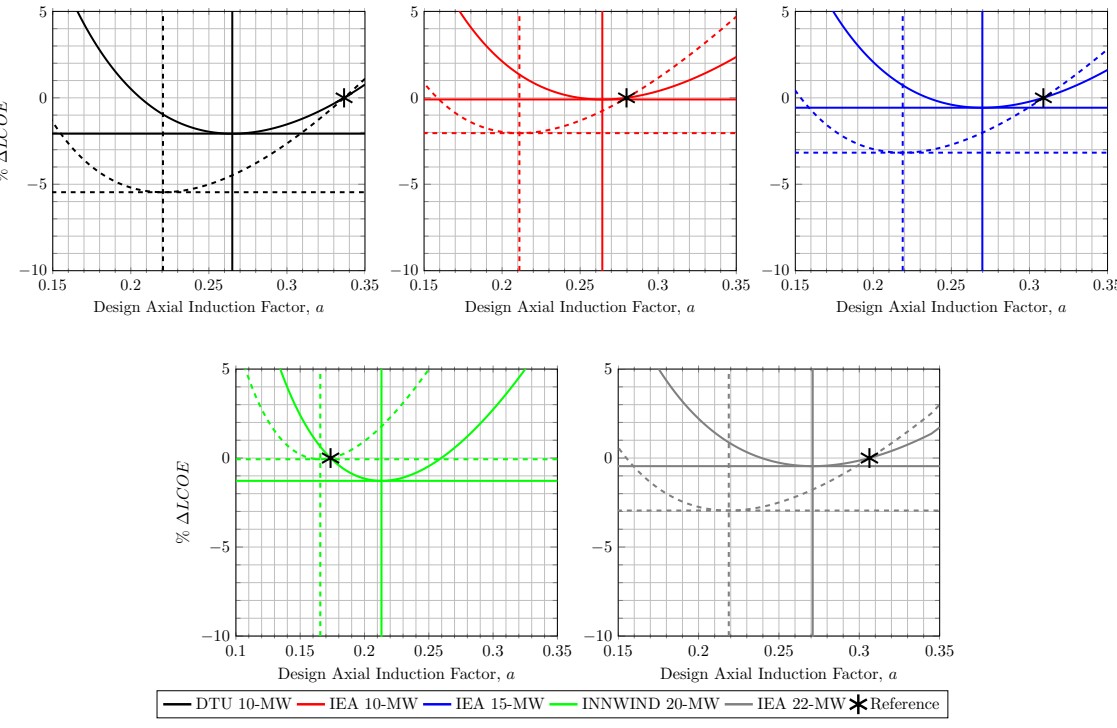

**Figure 13.** Range of optimum design axial induction factors at which to design an up-scaled version of the DTU 10-MW RWT (top-left), IEA 10-MW RWT (top-center), IEA 15-MW RWT (top-right), INNWIND 20-MW (bottom-left) and the IEA 22-MW RWT (bottom-right) for lowest LCOE.(Note rotor blade up-scaled from indicated reference point according to Eq. 4.)

Results for the DTU 10-MW RWT (Figure 13, top-left), were discussed in detail in the previous section. For the IEA 10-MW, the optimum range of design axial induction factors for a FACT blade design and potential LCOE reduction are slightly different than that of the DTU 10-MW (Figure 13, top-center). The upper and lower bounds of the optimum design axial





induction factor range from $a = 0.21$ to 0.255, with the highest possible reduction in LCOE around 3%. Possible improvement in LCOE is less than that of the DTU 10-MW. This is not necessarily a surprising result considering that the IEA 10-MW
is considered an optimized version of the DTU 10-MW. It is interesting to note that the conservative optimum design axial induction factor is near the mean design axial induction factor of the IEA 10-MW and gains in LCOE are predicted at this optimum are notably less than 1%. This shows that the IEA 10-MW is already designed near one LCOE optimum, however a second optimum potentially exists at $a = 0.21$ where an additional 3% can be gained with novel up-scaling.

Similar results are observed for the IEA 15- and 22-MW in Figures 13 top-right and bottom-right, respectively. The design
methodology and the tools used in the design of these rotors are similar to those of the IEA 10-MW (Bortolotti et al., 2019), thus it is also not surprising that the results for the IEA 15- and 22-MW are in the same range as those of the IEA 10-MW. The optimum design axial induction factor is between approximately $a = 0.22$ and 0.27 for both the IEA 15- and 22-MW. Again, the optimum determined from the old $\Delta CAPEX$ and $M_{rotor}$ relationships yields a design axial induction factor near the original design point of the reference rotor, which results in less than 1% gain in LCOE. Both of the predictions for the
IEA 15- and 22-MW based on the updated relationships developed in the present work could yield as much as a 2-3% LCOE reduction by lowering the design axial induction factor from 0.3 to 0.22.

A shift occurs in the trend when looking to the INNWIND 20-MW in Figure 13, bottom-left. Both reference 10-MW's and the 15-MW reference turbine have an optimum design point approximately centered around $a = 0.24$. The optimum design axial induction factor for an up-scaled INNWIND 20-MW shifts to lower values between $a = 0.165$ to 0.21. Compared to the
other turbines, the INNWIND 20-MW also has the smallest range for both the optimum design axial induction factor and LCOE with the estimate based on the relationships developed in this work giving only a 1% decrease, while the old relationships only give negligible reduction in LCOE for this reference turbine. The difference in design $a$-range and $\Delta LCOE$ identified by the routine for the INNWIND 20-MW compared to the IEA-series may be a result of different design methodologies and tools used in the detailed design process. Another interesting observation for this case is that the updated relationships developed in
this work suggest a reduction in blade length from the reference design to reduce cost of energy for this particular reference turbine.

## 5   Conclusions

It has been previously noted that tackling the grand challenges facing the future of wind energy will require new design methodologies, including solving the problem of increasing costs for up-scaled rotors. In an effort to address these grand
challenges, and using results from previous research conducted by the authors, the present work sought to identify a range of optimum design axial induction factors at which an up-scaled blade can be designed that strikes the right balance between increasing blade length to increase AEP while accounting for the additional cost and loading changes associated with longer blades. The purpose of identifying an ideal design axial induction factor is for subsequent use in inverse design; for example, Vortex Wake Method inverse design finds an optimum spanwise circulation distribution with variable axial induction factor





along the blade that integrates exactly to a thrust coefficient constraint resulting from the (average) design axial induction factor proposed in this work.

To investigate this hypothesis, an objective function for $\Delta LCOE$ as a function of design axial induction factor, $a$, was derived. In the process of developing the $\Delta LCOE$ objective function, new insights were gained for changes in CAPEX and OPEX with rotor up-scaling. As part of the CAPEX function development, new engineering approximations for rotor mass are
discussed that are suitable to large-diameter offshore wind turbines that use improved materials technologies and manufacturing processes. Additionally, a detailed OPEX model was developed using available data for real wind farms, and a relationship between turbine failure rate and damage-equivalent loads for failure-prone turbine subsystems was proposed.

Trends of $\Delta CAPEX(a)$, $\Delta OPEX(a)$, and $\Delta LCOE(a)$ were investigated, along with the sensitivity of these terms to the different relationships derived here, with the following conclusions:

– In general, CAPEX will always increase for an up-scaled rotor blade due to increased blade length and generator torque requirements, with as much as a 14% increase possible, while minimizing CAPEX favors designs near $a = 0.33$.

– Change in CAPEX relative to that of a reference rotor design may vary by as much as 5% depending on both the rotor mass approximation and distribution of CAPEX cost between the rotor, tower, and generator.

– Unlike $\Delta CAPEX$, $\Delta OPEX$ displays a local minimum because of the inverse relationship between planned and un-
planned maintenance costs and favors designs with lower axial induction factors near $a = 0.1$ where the reduction in $C_{PM}$ is highest.

– Change in OPEX is most sensitive to the assumed reliability of the turbine with the lowest OPEX reduction possible of only 1% for the highest failure rates compared to almost 12% for the lowest assumed failure rate; rotor mass and CAPEX cost distribution models can also have as much as a 6% impact on OPEX trends.

– As rotor diameter increases with decreasing design axial induction factor, Region II power production increases, leading to as much as a 3.5% increase in AEP in the case of an up-scaled DTU 10-MW RWT.

– When effects of CAPEX, OPEX, and AEP are all combined, $\Delta LCOE$ displays a clear minimum with axial induction factor; the highest possible LCOE reduction and optimum design axial induction factor do vary with combination of rotor mass prediction, capital cost distribution, and assumed failure rate of the turbine.

– When comparing the optimum design axial induction factor and $\Delta LCOE$ for a fatigue, aerodynamics, and cost-scaled (FACT) version of five different reference turbines, highest predicted $\Delta LCOE$ ranged from -1% to -5.5%, with the FACT blade using the DTU 10-MW blade as a reference having the highest possible gains.

– The optimum range of design axial induction factors as identified by the $\Delta LCOE(a)$ objective function falls somewhere between the low-induction rotor optimum ($a = 0.18$) and the Betz $C_P$-maximum optimum ($a = 0.33$).



Note that the $\Delta LCOE(a)$ function derived in the present work is only designed to give a starting point for an up-scaled rotor design; final LCOE results are subject to the detailed blade/rotor design process. Since all of the relationships derived here are for offshore turbines with rated power up to 22-MW, the authors also caution using this methodology to select an optimum axial induction factor at which to design an up-scaled rotor with a nameplate capacity of greater than 22-MW. For the rated power range covered in the present work, however, results demonstrate that it is possible to reduce LCOE of a given

reference rotor design by as much as 5% using the Fatigue, Aerodynamics, and Cost-scaled Turbine (FACT) design objective function derived here. LCOE savings achievable through implementation of the FACT objective function in future offshore wind turbine rotor designs will not only ensure wind remains cost-competitive in the energy market, but possibly further improve cost-competitiveness.

### Appendix A:  Nomenclature

**Roman**

$A$  $\Delta M_{rotor}$ proportionality coefficient

$a$  Axial-induction factor

$B$  $\Delta S_{tower}$ proportionality coefficient

$C$  $\Delta Q_{shaft}$ proportionality coefficient

$C_{CM}$  Cost of unplanned maintenance

$C_P$  Rotor power coefficient

$C_{PM}$  Cost of planned maintenance

$C_T$  Rotor thrust coefficient

$c$  Weibull shape factor

$c_{labor}$  Cost of labor

$c_r$  Cost of repair

$D$  "Other" CAPEX proportionality coefficient

$D_{LSS}$  Low-speed shaft diameter

$D_{pw}$  Pitch bearing ring diameter

$d$  Turbine downtime for repair





$E$   Change in fixed operations & maintenance cost proportionality coefficient

$F$   Change in cost of planned maintenance proportionality coefficient

$F_{B_x}$   Blade root out-of-plane force

$F_{B_y}$   Blade root in-plane force

$F_{B_z}$   Blade root axial force

$F_{LSS_x}$   Low-speed shaft axial force

$F_{LSS_y}$   Low-speed shaft lateral force

$F_{LSS_z}$   Low-speed shaft upward force

$F_T$   Rotor thrust force

$G$   Change in cost of unplanned maintenance proportionality coefficient

$H$   Tower height

$k$   Weibull function scale factor

$M_{B,rms}$   Root-mean-square blade root moment

$M_{Edge}$   Blade root in-plane edgewise moment

$M_{Flap}$   Blade root out-of-plane flap moment

$M_{LSS_y}$   Low-speed shaft up-down moment

$M_{LSS_z}$   Low-speed shaft side-side moment

$M_{rotor}$   Rotor mass

$MB_{Eq}$   Main bearing equivalent load

$OPEX_{Fixed}$   Fixed operations & maintenance cost

$OPEX_{Variable}$   Variable operations & maintenance cost

$P$   Wind turbine power

$\overline{P}$   Average wind turbine power

$P_{rated}$   Rated power





$PB_{Eq}$   Pitch bearing equivalent load

$p$   Probability density function

$p_r$   Proportion of downtime used for repair

$Q_{Aero}$   Rotor aerodynamic torque

$Q_{shaft}$   Shaft torque

$R$   Blade radius

R-squared   Coefficient of determination

$S_{tower}$   Tower structural capacity

$TSR$   Tip speed ratio

$u_{tip}$   Blade tip speed

$V$   Wind speed

$\overline{V}$   Average wind speed

$V_{hub}$   Hub-height wind speed

$V_{in}$   Cut-in wind speed

$V_o$   Design wind speed

$V_{out}$   Cut-out wind speed

$W$   Tower base width

$N$   Coefficient of generic $M_{rotor}$ function

**Greek**

$\Delta$   Relative change in parameter; up-scaled divided by reference

$\eta_{gen}$   Generator efficiency

$\rho$   Density

$\Omega$   Rotor rotational speed

$\lambda$   Failure rate

$\kappa$   $C_{PM}$ proportionality coefficients

$\gamma$   $C_{CM}$ proportionality coefficients



**Subscripts**

$i$  Individual turbine component index

$B$  Blade

$D$  Drivetrain

$G$  Generator

$P$  Pitch

**Superscripts**

$0$  Reference turbine value

$X$  Power law exponent

**Abbreviations**

AEP  Annual Energy Production

CAPEX  Capital expenditure

CIRCE  Centro de Investigacion de Recursos y Consumos Energéticos

CoF  Cost of Failure

LCOE  Levelized-Cost of Energy

OPEX  Operations & Maintenance Expenditure

DEL  Damage-equivalent load

VWM  Vortex wake method

LIR  Low-induction rotor

FACT  Fatigue, Aerodynamics, and Cost-scaled Turbine

DLC  Design load case

IEC  International Electrotechnical Commission

NTM  Normal Turbulence Model

DTU  Danish Technical University



IEA    International Energy Agency

NREL    National Renewable Energy Laboratory

LSS    Low-speed shaft

WISDEM    Wind-plant Integrated System Design and Engineering Model

LWK    Landwirtschaftskammer

Strath    University of Strathclyde

**Appendix B: O & M Reliability Data Analysis**

In this Appendix, the wind turbine reliability datasets from Dao et al. (2020) and Carroll et al. (2016) are provided, and additional details are given to show how the proportionality coefficients for $\Delta C_{PM}$, $\Delta C_{CM}$ and overall $\Delta OPEX$ are computed.

### B1   Dao et al. (2019) Reliability Data: CIRCE and LWK

After conducting a detailed review of wind turbine reliability data, Dao et al. (2020) identified three datasets to be the most comprehensive for the sub-assemblies of interest: CIRCE (Reder et al., 2016), LWK (Faulstich et al., 2011), and a dataset published by the University of Strathclyde. The dataset from the University of Strathclyde is from the work of Carroll et al. (2016) and will be detailed separately as it is a more complex dataset. The data provided in Dao et al. (2020) for CIRCE and LWK include failure rate, $\lambda_i$, and downtime, $d_i$, for six sub-assemblies: rotor blades, pitch system, drivetrain, generator, 845    converter, and electrical. Table B1 details the published information for both CIRCE and LWK.

**Table B1.** Failure rate and downtime data for the CIRCE and LWK databases, adapted from Dao et al. (2020).

|  | CIRCE | | LWK | |
| --- | --- | --- | --- | --- |
| Component | Failure Rate | Downtime [h] | Failure Rate | Downtime [h] |
| Rotor Blades | 0.044 | 190.73 | 0.194 | 42.12 |
| Pitch | 0.029 | 98.73 | 0.088 | 25.15 |
| Drivetrain | 0.016 | 165.85 | 0.030 | 118.19 |
| Generator | 0.029 | 320.64 | 0.140 | 74.36 |
| Converter | 0.006 | 74.17 | 0.053 | 29.87 |
| Electrical | 0.061 | 74.03 | 0.270 | 34.53 |



### B1.1 Planned Maintenance Cost

Using the failure rate data from Table B1, proportionality coefficients ($\gamma_i$'s in Eq. (21)) for the planned maintenance cost contribution are determined. To determine $\gamma_i$ for each subsystem, the failure rate of the individual subsystem is divided by the total number of failures, or $\lambda_i/\lambda$. The $\gamma_i$'s for CIRCE and LWK are given in Table B2.

**Table B2.** Proportion of total failures from relevant wind turbine subsystems from the CIRCE and LWK databases, adapted from Dao et al. (2020).

|  | CIRCE | | LWK | |
| --- | --- | --- | --- | --- |
| Component | Failure Rate | $\gamma_i$ | Failure Rate | $\gamma_i$ |
| Rotor Blades | 0.044 | 0.238 | 0.194 | 0.250 |
| Pitch | 0.029 | 0.157 | 0.088 | 0.113 |
| Drivetrain | 0.016 | 0.086 | 0.030 | 0.039 |
| Generator | 0.029 | 0.157 | 0.140 | 0.181 |
| Other | 0.067 | 0.362 | 0.323 | 0.417 |
| Total | 0.185 | - | 0.775 | - |

### B1.2 Unplanned Maintenance Cost

Using the failure rate and downtime data from Table B1, proportionality coefficients ($\kappa_i$'s in Eq. (22)) for the unplanned maintenance cost contribution are determined. To determine $\kappa_i$ for each subsystem, the unplanned maintenance cost for each subsystem, $C_{CM,i}$, is computed via

$$C_{CM,i} = \lambda_i \times (c_r + p_r \times d_i \times c_{labor}) \tag{B1}$$

For the CIRCE and LWK data in Dao et al. (2020), there is no reliability data given for repair cost, $c_r$, repair fraction, $p_r$, or cost of labor, $c_{labor}$. In a previous work using the same failure rate data to assess how changes in maintenance strategies impact OPEX cost (Dao et al., 2019), assumptions were made for these three variables that are listed in Table B3.

**Table B3.** $C_{CM}$ cost model assumptions taken from approximations by Dao et al. (2020)

| Parameter | Value | Unit |
| --- | --- | --- |
| Repair cost, $c_r$ | 30,000.00 | Euro |
| Labor cost, $c_{labor}$ | 300.00 | Euro/hour |
| Repair fraction, $p_r$ | 0.2 | |

Once $C_{CM,i}$ is computed for all subsystems, total $C_{CM}$ is found by summing up the individual $C_{CM,i}$'s and then $\kappa_i$ is found by dividing the individual unplanned maintenance cost by the total, or $C_{CM,i}/C_{CM}$. The $\kappa_i$'s for CIRCE and LWK are given





in Table B4. Note that the cost values in Table B3 are in Euros, and these values are multiplied by 1.1 to convert to USD for the final $C_{CM}$ calculation (as of May 2024).

**Table B4.** Calculated unplanned maintenance cost data, $C_{CM}$, for the CIRCE and LWK databases, adapted from Dao et al. (2020).

| Component | CIRCE | | | | LWK | | | |
|---|---|---|---|---|---|---|---|---|
| | $\lambda_i$ | $d_i$ | $C_{CM,i}$ | $\kappa_i$ | $\lambda_i$ | $d_i$ | $C_{CM,i}$ | $\kappa_i$ |
| | [failures/turbine] | [h] | [USD/yr] | [-] | [failures/turbine] | [h] | [USD/yr] | [-] |
| Rotor Blades | 0.044 | 190.73 | 2,005.88 | 0.250 | 0.194 | 42.12 | 6,941.34 | 0.244 |
| Pitch | 0.029 | 98.73 | 631.59 | 0.079 | 0.088 | 25.15 | 1,039.79 | 0.037 |
| Drivetrain | 0.016 | 165.85 | 1,274.43 | 0.159 | 0.030 | 118.19 | 5,712.03 | 0.201 |
| Generator | 0.029 | 320.64 | 1,570.70 | 0.196 | 0.140 | 74.36 | 3,335.89 | 0.117 |
| Converter | 0.006 | 74.17 | 227.37 | 0.028 | 0.053 | 29.87 | 1,853.47 | 0.065 |
| Electrical | 0.061 | 74.03 | 2,311.05 | 0.288 | 0.270 | 34.53 | 9,525.32 | 0.336 |
| **Total** | - | - | 8,021.01 | - | - | - | 28,407.85 | - |

## B2   Carroll et al. (2016) Reliability Data: Strath

The University of Strathclyde data used in Dao et al. (2020) is adapted from the work of Carroll et al. (2016). Rather than just reporting overall failure rate and downtime as was done in Dao et al. (2020), the original published reliability data in Carroll
et al. (2016) is broken down into type of repair required (minor repair, major repair, and major replacement) and even includes both repair cost and repair time for each subsystem and type of repair. The detailed dataset is given in Figure B1. This data is referred to at the "Strath" data in the manuscript.

### B2.1   Planned Maintenance Cost

Using the failure rate data from Figure B1(a), proportionality coefficients ($\gamma_i$'s in Eq. (21)) for the planned maintenance cost
contribution are determined and listed in Table B5. The $\gamma_i$'s for each subsystem are determined using the same method as those for the CIRCE and LWK data. Note that there are significantly more turbine subsystems listed in Figure B1(a) than in Table B5. As mentioned in Section 3.1, the only subsystems of interest to the present work are the blades, pitch, drivetrain, and generator. All other subsystems listed in Figure B1(a) are combined into the "Other" category.

### B2.2   Unplanned Maintenance Cost

To compute $C_{CM}$ for the CIRCE and LWK data, repair cost ($c_r$) and repair fraction ($p_r$) were constant for each component. In Carroll et al. (2016), however, the cost of repair and repair time ($t_r = p_r * d_i$) required to differ between a repair type and component, as seen in Figure B1(b) and (c), respectively. Using the repair cost (Figure B1b) and repair time (Figure B1c)





**(a) Failure rate [failures/turbine/year]**

| | Pitch / Hyd | Other Components | Generator | Gearbox | Blades | Grease / Oil / Cooling Liq. | Electrical Components | Contactor / Circuit Breaker / Relay | Controls | Safety | Sensors | Pumps/ Motors | Hub | Heaters / Coolers | Yaw System | Tower / Foundation | Power Supply / Converter | Service Items | Transformer |
|---|---|---|---|---|---|---|---|---|---|---|---|---|---|---|---|---|---|---|---|
| Major Replacement | 0.001 | 0.001 | 0.095 | 0.154 | 0.001 | 0.000 | 0.002 | 0.002 | 0.001 | 0.000 | 0.000 | 0.000 | 0.001 | 0.000 | 0.001 | 0.000 | 0.005 | 0.000 | 0.001 |
| Major Repair | 0.179 | 0.042 | 0.321 | 0.038 | 0.010 | 0.006 | 0.016 | 0.054 | 0.054 | 0.004 | 0.070 | 0.043 | 0.038 | 0.007 | 0.006 | 0.089 | 0.081 | 0.001 | 0.003 |
| Minor Repair | 0.824 | 0.812 | 0.485 | 0.395 | 0.456 | 0.407 | 0.358 | 0.326 | 0.355 | 0.373 | 0.247 | 0.278 | 0.182 | 0.190 | 0.162 | 0.092 | 0.076 | 0.108 | 0.052 |
| No Cost Data | 0.072 | 0.150 | 0.098 | 0.046 | 0.053 | 0.058 | 0.059 | 0.048 | 0.018 | 0.015 | 0.029 | 0.025 | 0.014 | 0.016 | 0.020 | 0.004 | 0.018 | 0.016 | 0.009 |

**(b) Repair Cost [Euros]**

| | Gearbox | Hub | Blades | Transformer | Generator | Power Supply / Converter | Contactor / Circuit Breaker / Relay | Pitch / Hyd | Yaw System | Controls | Electrical Components | Other Components | Sensors | Safety | Pumps/ Motors | Grease / Oil / Cooling Liq. | Heaters / Coolers | Service Items | Tower / Foundation |
|---|---|---|---|---|---|---|---|---|---|---|---|---|---|---|---|---|---|---|---|
| Minor Repair | 125 | 160 | 170 | 95 | 160 | 240 | 260 | 210 | 140 | 200 | 100 | 110 | 150 | 130 | 330 | 160 | 465 | 80 | 140 |
| Major Repair | 2500 | 1500 | 1500 | 2300 | 3500 | 5300 | 2300 | 1900 | 3000 | 2000 | 2000 | 2400 | 2500 | 2400 | 2000 | 2000 | 1300 | 1200 | 1100 |
| Major Replacement | 230000 | 95000 | 90000 | 70000 | 60000 | 13000 | 13500 | 14000 | 12500 | 13000 | 12000 | 10000 | 0 | 0 | 0 | 0 | 0 | 0 | 0 |

**(c) Repair Time [Hours]**

| | Hub | Blades | Gearbox | Contactor / Circuit Breaker / Relay | Generator | Power Supply / Converter | Yaw System | Other Components | Pitch / Hyd | Transformer | Controls | Electrical Components | Grease / Oil / Cooling Liq. | Heaters / Coolers | Sensors | Pumps/ Motors | Service Items | Tower / Foundation | Safety |
|---|---|---|---|---|---|---|---|---|---|---|---|---|---|---|---|---|---|---|---|
| No Cost Data | 8 | 28 | 7 | 5 | 13 | 10 | 9 | 8 | 17 | 19 | 17 | 7 | 3 | 5 | 8 | 7 | 9 | 6 | 2 |
| Minor Repair | 10 | 9 | 8 | 4 | 7 | 7 | 5 | 5 | 9 | 7 | 8 | 5 | 4 | 5 | 8 | 4 | 7 | 5 | 2 |
| Major Repair | 40 | 21 | 22 | 19 | 24 | 14 | 20 | 21 | 19 | 26 | 14 | 14 | 18 | 14 | 6 | 10 | 2 | 7 | 7 |
| Major Replacement | 298 | 288 | 231 | 150 | 81 | 57 | 49 | 36 | 25 | 1 | 12 | 18 | 0 | 0 | 0 | 0 | 0 | 0 | 0 |

**Figure B1.** (a) Failure rate, (b) repair cost, and (c) repair time reliability data adapted from Carroll et al. (2016).





**Table B5.** Proportion of total failures from relevant wind turbine subsystems for the Strath reliability database, adapted from Carroll et al. (2016).

| $\lambda$ | Blades | Pitch | Drivetrain | Generator | Other | **Total** |
|---|---|---|---|---|---|---|
| Minor Repair | 0.456 | 0.824 | 0.395 | 0.485 | 0.014 | - |
| Major Repair | 0.010 | 0.179 | 0.038 | 0.321 | 0.514 | - |
| Major Replacement | 0.001 | 0.001 | 0.154 | 0.095 | 4.018 | - |
| No Cost Data | 0.053 | 0.072 | 0.046 | 0.098 | 0.499 | - |
| **Total** | 0.520 | 1.076 | 0.633 | 0.999 | 5.045 | 8.273 |
| $\gamma_i$ | 0.063 | 0.130 | 0.077 | 0.121 | 0.610 | - |

reliability data, the individual $C_{CM,i}$ for each component and repair type are computed with the slightly adjusted formula for $C_{CM_i}$:

$$C_{CM_i} = \lambda_i \times (c_{r_i} + t_{r_i} \times c_{labor}) \tag{B2}$$

Note that cost of labor ($c_{labor}$) is still constant at 300 Euros/hour as this information was not provided in the comprehensive Strathclyde dataset. Again, only the unplanned maintenance cost data for the blades, pitch, drivetrain, and generator are presented here. The individual $C_{CM}$ cost for all other components and each repair type are computed and then the costs are summed up within each repair type to come up with $C_{CM}$ for minor repairs, major repairs, and major replacements for all "other" subsystems. This method of handling the "other" components accurately accounts for the variation in failure rate, repair cost, and repair time for all components and repair types. Note that the cost values in Table B6 are in USD after the Euro cost data are multiplied by 1.1 to convert to USD for the final $C_{CM}$ calculation (as of May 2024).

**Table B6.** $C_{CM}$ data from the Strath database, adapted from Carroll et al. (2016)

| Component | Minor Repair [Euros/yr] | Major Repair [Euros/yr] | Major Replacement [Euros/yr] | $C_{CM}$ [Euros/yr] | $\kappa_i$ [-] |
|---|---|---|---|---|---|
| Blades | 1,439.59 | 85.80 | 194.04 | 1,719.43 | 0.007 |
| Pitch | 2,637.62 | 1,496.44 | 23.65 | 4,157.71 | 0.016 |
| Drivetrain | 1,097.11 | 380.38 | 50,701.42 | 52,178.91 | 0.199 |
| Generator | 1,205.71 | 3,778.17 | 8,809.35 | 13,793.23 | 0.053 |
| Other | 126,317.88 | 56,992.32 | 6,549.62 | 189,859.82 | 0.725 |
| **Total** | 132,697.92 | 62,733.11 | 66,278.08 | 261,709.11 | - |



## B3    Total OPEX Coefficients

To compute the proportionality coefficients $E$, $F$, and $G$ in Eq. (20) for the relative contributions of each OPEX cost component
to total OPEX, estimates for $C_{CM}$ from the previous sections are combined with assumed values for $OPEX_{Fixed}$ and $C_{PM}^0$.
In Dao et al. (2019), $OPEX_{Fixed}$ is assumed to be 10% of CAPEX and the baseline value of $C_{PM}^0$ is 15% of CAPEX. To
estimate CAPEX, the DTU 10-MW RWT is used as the baseline rotor from which CAPEX is estimated using WISDEM. Using
this CAPEX value, $OPEX_{Fixed}$ and $C_{PM}^0$ are computed, then total OPEX is computed according to Eq. (14). Cost results
for each component are given in Table B7. Dividing the $OPEX_{Fixed}$, $C_{PM}^0$, and $C_{CM}$ columns by the total OPEX yields the
proportional contribution of each term to the total OPEX cost. These constants are also listed next to their respective component
in Table B7 below.

**Table B7.** Cost estimates for the individual components of the OPEX equation and estimated total OPEX using the DTU 10-MW as the
reference turbine.

| Data | $OPEX_{Fixed}$ [USD] | $E$ [-] | $C_{PM}^0$ [USD] | $F$ [-] | $C_{CM}$ [USD] | $G$ [-] | OPEX [USD] |
|---|---|---|---|---|---|---|---|
| CIRCE | 1,368,657 | 0.40 | 2,052,986 | 0.598 | 8,021 | 0.002 | 3,449,663 |
| LWK | 1,368,657 | 0.40 | 2,052,986 | 0.59 | 28,408 | 0.01 | 3,450,049 |
| Strath | 1,368,657 | 0.37 | 2,052,986 | 0.56 | 261,709 | 0.07 | 3,683,351 |

*Author contributions.* D. Major conducted the research and wrote the manuscript. S. Schmitz proposed and reviewed the present work.

*Competing interests.* The authors declare that they have no conflict of interest.

*Acknowledgements.* The authors would like to thank Jason Jonkman, Garrett Barter, Shaun Sheng, and Jonathan Keller at the National
Renewable Laboratory (NREL) for their insights into component failure, the development of IEA-series turbine models, and for providing
knowledge on how to use several of the data analysis tools used to conduct this work.





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
