# Peer review of "The optimum range of design axial induction factors for lowest levelized-cost of energy"

_Wind Energy Science, 2024_

## Referee Comment (RC1)

The paper presents a methodology to select the optimal induction coefficient of a large offshore rotor design. The optimal coefficient is based on LCOE minimization. Detailed relations for AEP and especially CAPEX and OPEX are developed in the paper with reference to previous studies and existing databases. The outcomes are interesting and in-line with the existing literature on the topic. I generally find the manuscript to be well written and the topic to be relevant and well presented. Some comments for the authors below

- My main concern is related to the assumptions made on tower cost. Rotor radius can increase more than 30% at a=0.1. (Fig 1) With some of the suggested optimum inductions (a=0.2-0.25) increases are between 5 and 10%. Offshore turbines typically feature "stubby" towers, often leaving a clearance between blade tip and sea surface of about 30m, which is typically the accepted minimum. With the size of current offshore rotors they would probably need to be taller if a 10% increase in radius is desired. This would impact CAPEX

- The study deals with offshore turbines, is the foundation cost accounted for in ΔLCOE? (monopile/jacket or floater)

- L42: *"As more wind farms move offshore, wind turbine size can also increase due to less stringent noise requirements"* - Can you provide references for the statement? Noise should be more strongly dependent on tip speed rather than rotor size

- CAPEX is based on thrust coefficient, which appears to be related to ultimate loads. Fatigue is only part of OPEX. Nevertheless, despite positively influencing only OPEX, the results show significant LCOE reduction with induction factor, which is significant. I am also aware that the impact of induction coefficient on ultimate loads is hard to estimate as these depend on man factors, not least the control strategy. However, I would encourage authors to comment on this.

- In relation to the previous comment, many quantities in the study are related to rotor radius (blade mass & capex for instance). Often, they are found through data fitting of existing turbine models of different radii. These turbines typically also feature higher power (as radius increases) and also higher induction coefficient than what this study leans towards. How is this expected to impact results?

- Eq. 30: the assumption here is that the rotor is an ideal rotor according to momentum theory. At a=0.33 the rotor is operating at the Betz limit. Unless I have missed something, this means that power and AEP are overestimated in the current study

- Fig. 7: which mass relation is used for the rotor blades here?

---

## Referee Comment (RC2)

One has:

$$\Delta LCOE(a) = \frac{LCOE(a)}{LCOE_0}$$

We assume that the following formula holds:

$$LCOE(a) = \frac{Costs(a)}{AEP(a)} = \frac{CAPEX(a) + OPEX(a)}{AEP(a)}$$

Then, one has:

$$\Delta LCOE(a) = \frac{\dfrac{Costs(a)}{Costs_0}}{\dfrac{AEP(a)}{AEP_0}} = \frac{\Delta Costs(a)}{\Delta AEP(a)}$$

If we expand $\Delta Costs(a)$, we obtain:

$$\Delta Costs(a) = \frac{CAPEX(a)}{Costs_0} + \frac{OPEX(a)}{Costs_0} \neq \Delta CAPEX(a) + \Delta OPEX(a)$$

Therefore, the proposed formula for $\Delta LCOE$ is not valid

$$\Delta LCOE(a) \neq \frac{\Delta CAPEX(a) + \Delta OPEX(a)}{\Delta AEP(a)}$$